# DISCO: Mitigating Bias in Deep Learning with Conditional DIStance COrrelation

## Abstract

Dataset bias often leads deep learning models to exploit spurious correlations instead of task-relevant signals. We introduce the Standard Anti-Causal Model (SAM), a unifying causal framework that characterizes bias mechanisms and yields a conditional independence criterion for causal stability. Building on this theory, we propose $DISCO_m$ and sDISCO, efficient and scalable estimators of conditional distance correlation that enable independence regularization in gradient-based models. Across six diverse datasets, our methods consistently outperform or are competitive in existing bias mitigation approaches, while requiring fewer hyperparameters and scaling seamlessly to multi-bias scenarios. This work bridges causal theory and practical deep learning, providing both a principled foundation and effective tools for robust prediction. Source Code: `https://github.com/***`.

## 1 Introduction

Dataset bias poses a persistent challenge for modern (deep) learning methods (Jones et al., 2023), as it can induce spurious shortcuts that fail to generalize or obscure the task-relevant signal (Geirhos et al., 2020). In Alzheimer's disease prediction, for example, age often acts as a confounder in a fork structure (see Fig. 1a). If left unaddressed, the prediction system may rely predominantly on age-related features rather than identifying disease markers or their interaction effects (Zhao et al., 2020). Another widespread phenomenon is collider bias (often referred to as sampling bias in this context), illustrated in Fig. 1b. Collider bias typically arises implicitly through data collection and creates spurious associations between variables that are not causally related. For instance, hospitalization can function as a hidden collider that connects risk factors and diseases without an underlying causal mechanism (Griffith et al., 2020). A third source of dataset bias arises from mediator variables (Fig. 1c), which may carry information that is irrelevant or unstable for the prediction task. Following (Shpitser & VanderWeele, 2011; Pearl, 2012), we focus on the direct effects ($DE$) of the target variable and consider indirect effects ($IE$) through such mediators as undesirable shortcuts.

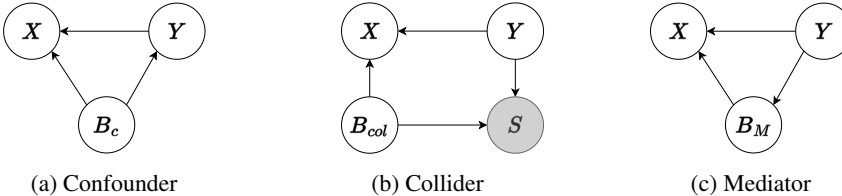

|          |          |          |
|----------|----------|----------|
| (a) Confounder | (b) Collider | (c) Mediator |

Figure 1: Canonical causal structures that induce dataset bias. Grey nodes indicate conditioning.

While such a pathway specific analysis is heavily used in statistics and causal inference, this point of view is neglected in the deep learning community. We show that this line of thought and analysis is equally important and useful in deep learning prediction settings. Therefore, we first introduce the *Standard Anti-Causal Model (SAM)*, a unifying framework that characterizes bias mechanisms in prediction tasks. To be more precise, it assumes an *anti-causal prediction setup* (Schölkopf et al., 2012), where the target variable is assumed to generate the input data (Fig. 1), and enables precise reasoning about counterfactual pathways. SAM not only allows us to derive a clear conditional

independence criterion for *causal stability*, that describes a predictors stability against counterfactual changes in bias attributes, but also serves as a general analytic tool for studying counterfactual bias effects when the full data-generating process is under control.

While prior work has proposed tailored methods to address confounder bias (Neto, 2020; Zhao et al., 2020) or collider bias (Darlow et al., 2020), we deduce from our framework that the specific causal nature of the bias is irrelevant: all types can be mitigated uniformly, provided a stable direct effect of interest exists.

Specifically, we show that when no unobserved backdoor paths exist between input $X$ and target $Y$, the conditional independence criterion

$$\hat{Y} \perp \mathbf{B} \mid Y, \tag{1}$$

where $\mathbf{B}$ denotes all observed biases and $\hat{Y}$ the model output, ensures *causal stability*. In this case, all counterfactual indirect and spurious effects ($SE$) vanish, and predictions rely solely on the direct causal path from $Y$ to $\hat{Y}$. When unobserved backdoor paths are present, causal stability can no longer be guaranteed; however, the conditional independence criterion remains valuable, as it blocks all biased paths involving $\mathbf{B}$. Similar independence-based criteria have been used empirically in fairness and bias mitigation (Makar & D'Amour, 2022; Kaur et al., 2022; Puli et al., 2021), and our causal pathway analysis provides a theoretical foundation for their effectiveness. Related approaches, such as (Quinzan et al., 2022), arrive at partially overlapping conclusions but without leveraging causal graphical analysis. Compared to (Veitch et al., 2021), who prove counterfactual invariance under more restrictive assumptions, our results generalize to richer anti-causal prediction settings.

Moreover, building on this theoretical foundation, we develop a novel practical approach for enforcing causal stability in black-box models. While conditional distance correlation has been proposed before (Wang et al., 2015), its existing formulations are computationally prohibitive and essentially unusable in modern optimization and deep learning. We resolve this by introducing two new estimators, compatible with backpropagation: $\text{DISCO}_m$ and its single-shot variant sDISCO. Unlike other shortcut-removal methods, our approach supports arbitrary combinations of target and bias variable types, scales effectively to multivariate settings, and requires significantly fewer hyperparameter choices. Therefore, we enable for conditional regularization of deep learning models.

In Section 4, we demonstrate across six diverse datasets that $\text{DISCO}_m$ and sDISCO consistently outperform existing bias mitigation methods, while having no scaling issues to multi-bias scenarios. Beyond empirical performance, we show how SAM further enables fine-grained pathway analysis of decision processes, highlighting the dual theoretical and practical impact of our work.

## 1.1 Contextualization in CRL and Bias Mitigation

**Causal Representation Learning (CRL).** Our work aligns with the broader goals of Causal Representation Learning (Schölkopf et al., 2021), specifically the objective of learning representations that are robust and generalize across environments. A full contextualization within CRL can be read in Appendix B.

**Bias/Shortcut Mitigation Positioning.** While bias mitigation is an established field, our framework provides a structural foundation that generalizes prior findings. See appendix at B for a full discussion how our work relates to Makar & D'Amour (2022); Veitch et al. (2021); Puli et al. (2021).

## 2 Causal Invariance

### 2.1 Standard Anti-Causal Model (SAM) and Causal Stability

Our Standard Anti-Causal Model (SAM) is inspired by the work of (Plecko & Bareinboim, 2022), and we therefore adopt notation closely aligned with theirs. We define structural causal models (SCMs) as quadruples $(\mathbf{V}, \mathcal{F}, \mathbf{U}, P_\mathbf{U})$, where $\mathbf{U}$ are the exogenous variables with distribution $P_\mathbf{U}$ defined outside the model. The set $\mathbf{V}$ contains the endogenous random variables, which are generated through deterministic functions of their parents in $\mathbf{V}$ combined with exogenous noise from $\mathbf{U}$. Specifically, $\mathcal{F} = \{f_1, \ldots, f_n\}$ is a set of functions such that $v_i \leftarrow f_i(pa(V_i), U_i)$, where $pa(V_i) \subseteq \mathbf{V}$ are the parents of $V_i$, and $U_i \in \mathbf{U}$ is the exogenous variable associated with $V_i$.

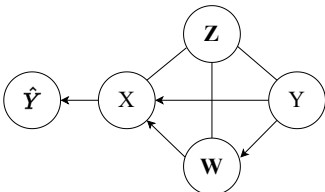

Figure 2: SAM graph. $Y$ is the target, $\mathbf{Z}$ are variables on active, non-directed paths between $X$ and $Y$, and $\mathbf{W}$ are mediator variables assumed to be unwanted shortcuts. $\hat{Y}$ denotes the prediction from a fixed prediction model. See appendix Fig. C.1 for more details on permitted relationships.

Figure 2 illustrates the SAM considered in this work. It represents a prototypical causal structure that encompasses many relevant scenarios in anti-causal prediction. We partition the variables $\mathbf{V}$ into three groups: the target $Y \in \mathbf{V}$, backdoor variables $\mathbf{Z} \subset \mathbf{V}$ (confounders and variables on open collider paths), and mediator variables $\mathbf{W} \subset \mathbf{V}$. For simplicity, independent exogenous noise variables $\mathbf{U}$, and variables $\mathbf{V}' \subset \mathbf{V}$ that do not open paths between our input $X$ and our target $Y$, are omitted from the figure. While our analysis extends to multidimensional targets, we restrict to a scalar $Y$ for clarity.

For counterfactual and interventional reasoning, we adopt the notation of (Pearl, 2009) and (Plecko & Bareinboim, 2022). The expression $P(X_w)$ is equivalent to $P(X \mid do(W = w))$ at the SCM level, while $P(X_w \mid W = w')$ denotes a genuine counterfactual statement. At the unit level, counterfactuals are defined for fixed exogenous realizations $\mathbf{U} = \mathbf{u}$ and denoted $X_w(\mathbf{u})$. These are deterministic functions rather than random variables and are generally non-identifiable. In the following, we assume that SAM holds for our prediction task; later, we discuss violations of these assumptions and their implications.

Throughout our definitions and proofs, we use summation (implicitly assuming the counting measure) for discrete random variables. With appropriate measure-theoretic extensions, these results carry over to continuous variables by replacing sums with integrals.

Our main goal for this section is to establish why the conditional independence criterion as presented in the introduction, reading $\hat{Y} \perp \mathbf{B} \mid Y$, is sufficient to mitigate bias and directly derive this by causal reasoning.

**Measuring Effects of $Y$ on $\hat{Y}$.** A naive way to quantify the influence of $Y$ on $\hat{Y}$ is via the total variation ($TV$) as defined in (Plecko & Bareinboim, 2022):

$$TV_{y_0,y_1}(\hat{y}) = P(\hat{y} \mid y_1) - P(\hat{y} \mid y_0), \tag{2}$$

where lowercase letters denote realizations $\hat{y}, y_1$, and $y_0$. Classical maximum likelihood estimation implicitly maximizes this difference. Assuming binary classification for simplicity, maximizing $P(\hat{Y} = y \mid Y = y)$ corresponds to maximizing the True Positive Rate and minimizing the False Positive Rate, thereby maximizing their difference (TV). However, in the SAM graph, we see that $TV$ captures all types of dependencies, including shortcuts and spurious associations over the pathways $Y$—$\mathbf{Z}$—$\hat{Y}$ and $Y \to \mathbf{W} \to \hat{Y}$, rather than only the direct causal influence $Y \to \hat{Y}$.

**Counterfactual-(DE, IE, SE)** (Plecko & Bareinboim, 2022). SCMs allow us to specify, and in some cases identify, path-specific effects. Given observations $Y = y$, $\mathbf{Z} = \mathbf{z}$, and $\mathbf{W} = \mathbf{w}$, we define the counterfactual direct (ctf-DE) and indirect (ctf-IE) effects as

$$ctf\text{-}DE_{y_0,y_1}(\hat{y} \mid y, \mathbf{w}, \mathbf{z}) = P(\hat{y}_{y_1,w_{y_0}} \mid y, \mathbf{w}, \mathbf{z}) - P(\hat{y}_{y_0} \mid y, \mathbf{w}, \mathbf{z}), \tag{3}$$

$$ctf\text{-}IE_{y_1,y_0}(\hat{y} \mid y, \mathbf{w}, \mathbf{z}) = P(\hat{y}_{y_1,w_{y_0}} \mid y, \mathbf{w}, \mathbf{z}) - P(\hat{y}_{y_1} \mid y, \mathbf{w}, \mathbf{z}). \tag{4}$$

Additionally, we define the counterfactual spurious effect (*ctf-SE*) as

$$ctf\text{-}SE_{y_1,y_0}(\hat{y}) = P(\hat{y}_{y_1} \mid y_0) - P(\hat{y}_{y_1} \mid y_1), \tag{5}$$

which captures all non-directed dependencies between $Y$ and $\hat{Y}$.

For notational/mathematical simplicity, without losing any validity of the below statements, we omit the input variable $X$ and treat paths through $X$ as directly reaching $\hat{Y}$.

**Proposition 1** (TV-Decomposition). *The total variation can be decomposed into direct, indirect, and spurious components as*

$$TV_{y_0,y_1}(\hat{y}) = \sum_{\mathbf{w},\mathbf{z}} \Big[ (ctf\text{-}DE_{y_0,y_1}(\hat{y} \mid y, \mathbf{w}, \mathbf{z}) - ctf\text{-}IE_{y_1,y_0}(\hat{y} \mid y, \mathbf{w}, \mathbf{z}))P(\mathbf{w}, \mathbf{z} \mid y) \Big]$$
$$- ctf\text{-}SE_{y_1,y_0}(\hat{y}). \tag{6}$$

*The proof is given in Appendix D.1.*

We assume that the only stable relationship under distribution, covariate, or domain shifts is the direct effect $ctf\text{-}DE_{y_0,y_1}$ (i.e., $Y \to X \to \hat{Y}$). We now identify an observational criterion that ensures both $ctf\text{-}IE_{y_1,y_0}$ and $ctf\text{-}SE_{y_1,y_0}$ vanish.

**Definition 1** (Causal Stability). *A predictor is* causally stable *if for all relevant variables and realizations it satisfies*

$$ctf\text{-}IE_{y_0,y_1}(\hat{y} \mid y, \mathbf{w}, \mathbf{z}) = 0 \quad and \quad ctf\text{-}SE_{y_1,y_0}(\hat{y}) = 0. \tag{7}$$

**Theorem 1** (Cond. Independence $\Rightarrow$ Causal Stability). *If a model's predictions satisfy $\hat{Y} \perp \mathbf{W}, \mathbf{Z} \mid Y$ under SAM, then the model is* causally stable. *The proof is provided in Appendix D.2.*

Note that causal stability alone does not guarantee predictive usefulness. For example, a constant predictor that outputs the same $\hat{Y}$ irrespective of the input trivially satisfies causal stability, but has no meaningful direct effect. The following result shows how to obtain both stability and strong predictive signal.

**Theorem 2** (Maximizing $ctf\text{-}DE_{y_0,y_1}$). *A maximum likelihood (MLE) predictor that satisfies $\hat{Y} \perp \mathbf{W}, \mathbf{Z} \mid Y$ also maximizes the direct effect $ctf\text{-}DE_{y_0,y_1}$. The proof is provided in Appendix D.2.*

**Corollary 1** (Indirect vs. Spurious Effects). *There is no need to distinguish between mediated and spurious bias effects when analyzing causal stability. By defining the set of all bias variables as $\mathbf{B} = \mathbf{W} \cup \mathbf{Z}$, the conditional independence criterion reduces to $\hat{Y} \perp \mathbf{B} \mid Y$.*

Thus, to obtain a stable predictor that relies solely on direct effects from $Y$ to $X$, we can formulate the following constrained optimization problem:

$$\min_{\theta} \quad \mathbb{E}_{(X,Y)} \left[ L\left(Y, g_\theta(X)\right) \right]$$
$$\text{s.t.} \quad \hat{Y} \perp \mathbf{B} \mid Y, \tag{8}$$

where $\hat{Y} = g_\theta(X)$, $g$ is a learning model parametrized by $\theta$, and $L$ is a standard loss function (e.g., mean squared error, cross-entropy) corresponding to an MLE under appropriate noise assumptions.

## 2.2 ANALYZING ASSUMPTIONS OF SAM

**Sufficient Variability (Positivity).** A fundamental requirement for our method, and indeed for any observational bias mitigation approach, is the assumption of sufficient variability or *overlap*. Analogous to the positivity assumption in causal inference, we assume that for any target realization $y$, the data distribution has support over the bias attributes $b$, i.e., $P(B = b \mid Y = y) > 0$. For example, to disentangle the spurious correlation between 'waterbird' and 'water background', the dataset must contain, however rarely, counter-examples such as waterbirds in land environments. Consequently, if $B$ is a deterministic function of $Y$ (zero overlap), the confounding is non-identifiable from observational data alone and debiasing the predictions is generally impossible without further guidance. This limitation applies to all bias mitigation methods.

**Selective Mediation.** While our framework treats mediators as sources of bias by default, practical applications often involve mediators that carry robust, task-relevant causal signals. In such cases, we can distinguish between unstable mediators and a subset of useful mediators $\mathbf{W}_{stable} \subset \mathbf{W}$. We then redefine the regularization bias set as $\mathbf{B}' = (\mathbf{W} \setminus \mathbf{W}_{stable}) \cup \mathbf{Z}$. Our theoretical results (Theorem 1) transfer directly to this setting: the model is regularized to be independent of $\mathbf{Z}$ and unstable mediators, while remaining free to utilize information from $\mathbf{W}_{stable}$.

**Unobserved Confounding.** Finally, an important violation of our assumptions arises from unobserved pathways of information. Since confounders and other spurious effects are modeled by $\mathbf{Z} \subseteq \mathbf{B}$,

the absence of some confounders means we work with an incomplete set of observations. The true set of spurious attributes is then $\mathbf{Z}_{true} = \mathbf{Z} \cup \mathbf{Z}'$, where $\mathbf{Z}'$ represents unobserved confounders. In this case, even an MLE predictor constrained by $\hat{Y} \perp \mathbf{B} \mid Y$ remains biased, as information may still leak through unobserved paths involving $\mathbf{Z}'$. Nonetheless, it is straightforward to show that all observed paths involving $\mathbf{B}$ remain blocked, and the criterion $\hat{Y} \perp \mathbf{B} \mid Y$ remains the best achievable approximation of causal stability. We verify this experimentally in Section 4.

## 3 DISTANCE CORRELATION FOR CONDITIONAL INDEPENDENCE

In this section, our main goal is to establish a method that enables black-box predictors like neural networks to attain causally stable solutions by optimizing towards the conditional independence criterion in Eq. 8.

Distance correlation (Székely et al., 2007), provides a powerful tool for quantifying statistical dependence between two arbitrary random vectors of potentially different dimensionalities. Subsequent work (Póczos & Schneider, 2012; Wang et al., 2015; Pan et al., 2017) extended this concept to the conditional setting, where dependence is measured given an additional random vector. However, existing implementations are computationally expensive and largely impractical for modern deep learning and optimization.

We address this gap by introducing estimators of conditional distance correlation, which we call $\mathrm{DISCO}_m$ and sDISCO. $\mathrm{DISCO}_m$ measures conditional dependence between two arbitrary random vectors given a third one. We further propose an even more efficient variant, sDISCO, which reduces the computation to a single-step operation. We prove that, under suitable conditions, both $\mathrm{DISCO}_m$ and sDISCO vanish if and only if the variables are conditionally independent. Finally, we show how these estimators can be integrated into deep learning models to optimize for causal stability.

### 3.1 THEORY

Let $(\mathcal{X}, d_{\mathcal{X}})$, $(\mathcal{Y}, d_{\mathcal{Y}})$, and $(\mathcal{Z}, d_{\mathcal{Z}})$ be metric spaces. Let $(X, Y, Z)$ be random elements defined on a common probability space, taking values in $\mathcal{X} \times \mathcal{Y} \times \mathcal{Z}$, with joint distribution $P$. Denote by $P_{(X,Y)|Z}$ the regular conditional distribution of $(X, Y)$ given $Z$. For each $z \in \mathcal{Z}$, let $\theta_z := P_{(X,Y)|Z=z}$ be a Borel probability measure on $\mathcal{X} \times \mathcal{Y}$.

**Definition 2** (Finite first moments). *We say that $P_{(X,Y)|Z=z}$ has finite first moments if*

$$\int d_{\mathcal{X}}(x, o_{\mathcal{X}}) + d_{\mathcal{Y}}(y, o_{\mathcal{Y}}) \, d\theta_z(x, y) < \infty \tag{9}$$

*for some (and hence all, by the triangle inequality) base points $o_{\mathcal{X}} \in \mathcal{X}$ and $o_{\mathcal{Y}} \in \mathcal{Y}$.*

**Definition 3** (Conditional distance covariance). *Let $P_Z$ denote the distribution of $Z$. For $z \in \mathcal{Z}$, let $\mu_z := P_{X|Z=z}$ and $\nu_z := P_{Y|Z=z}$ be the conditional marginals. For $\theta_z$, define the distance-centered kernels*

$$d_{\mu_z}(x, x') := d_{\mathcal{X}}(x, x') - a_{\mu_z}(x) - a_{\mu_z}(x') + D(\mu_z), \tag{10}$$

*where $a_{\mu_z}(x) := \int d_{\mathcal{X}}(x, x') \, d\mu_z(x')$ and $D(\mu_z) := \int\int d_{\mathcal{X}}(x, x') \, d\mu_z(x) d\mu_z(x')$. The definition of $d_{\nu_z}$ is analogous.*

*The conditional distance covariance is then given by*

$$\mathrm{dCov}^2(X, Y \mid Z) := \mathbb{E}_Z \left[ \mathrm{dCov}^2(X, Y \mid Z = z) \right], \tag{11}$$

*with*

$$\mathrm{dCov}^2(X, Y \mid Z = z) := \int d_{\mu_z}(x, x') \, d_{\nu_z}(y, y') \, d\theta_z(x, y) \, d\theta_z(x', y'). \tag{12}$$

**Theorem 3** (Conditional independence and strong negative type). *Assume $\mu_z$, $\nu_z$, and $\theta_z$ have finite first moments. Further suppose that the metric spaces $\mathcal{X}$ and $\mathcal{Y}$ are of strong negative type[1]. Then*

$$\mathrm{dCov}^2(X, Y \mid Z) = 0 \iff P_{(X,Y)|Z=z} = P_{X|Z=z} \otimes P_{Y|Z=z} \quad \text{for } P_Z\text{-almost every } z. \tag{13}$$

*That is, $X$ and $Y$ are conditionally independent given $Z$ almost surely. A proof, together with full definitions and necessary lemmas, is provided in Appendix E.*

---

[1] Strong negative type ensures that distance-based measures such as distance covariance uniquely identify independence; see Appendix E.

Since distance covariance is unbounded, we instead use its correlation analogue, which lies in $[0, 1]$ and is more interpretable.

**Definition 4** (Conditional distance correlation)**.** *The* conditional distance correlation *between $X$ and $Y$ given $Z$ is defined as*

$$\text{dCor}^2(X, Y \mid Z) := \frac{\text{dCov}^2(X, Y \mid Z)}{\sqrt{\text{dCov}^2(X, X \mid Z) \ \text{dCov}^2(Y, Y \mid Z)}}, \tag{14}$$

*with the convention $0/0 := 0$.*

### 3.2 Sample Estimation

Let $\{(X_i, Y_i, Z_i)\}_{i=1}^n$ be an i.i.d. sample, where $X, Y, Z$ are random vectors in Euclidean spaces, as is common in deep learning. We use the Euclidean distance $d(x, x') = \|x - x'\|$, which is of strong negative type (see, e.g., (Sejdinovic et al., 2013)).

First, fix a positive-definite kernel $K_h : \mathcal{Z} \times \mathcal{Z} \to \mathbb{R}_{\geq 0}$ with bandwidth $h$ (e.g., an RBF kernel), and define the kernel weight matrix $W = [w_{ij}]$ by

$$w_{ij} = K_h(Z_i, Z_j), \quad w_{i.} = \sum_j w_{ij}, \quad w_{.j} = \sum_i w_{ij}, \quad S = \sum_{i,j} w_{ij}. \tag{15}$$

Next, compute the pairwise distance matrices $A = [a_{ij}]$, $B = [b_{ij}]$ with

$$a_{ij} = d_{\mathcal{X}}(X_i, X_j), \quad b_{ij} = d_{\mathcal{Y}}(Y_i, Y_j). \tag{16}$$

**Method 1: DISCO$_m$.** For each $i$, define row-normalized weights $w_k^{(i)} = w_{ik}/w_{i.}$. Construct locally centered matrices:

$$A_{k\ell}^{(i)} = a_{k\ell} - \sum_k^n w_k^{(i)} a_{k\ell} - \sum_\ell^n w_\ell^{(i)} a_{k\ell} + \sum_{k,\ell}^n w_k^{(i)} w_\ell^{(i)} a_{k\ell}, \tag{17}$$

and define $B_{k\ell}^{(i)}$ analogously. The local conditional squared distance covariance at $Z_i$ is

$$\widehat{\text{dCov}}_{DISCO}(X, Y \mid Z_i) = \frac{1}{\sum_{k,\ell}^n w_{ik} w_{i\ell}} \sum_{k,\ell}^n w_{ik} w_{i\ell} A_{k\ell}^{(i)} B_{k\ell}^{(i)}. \tag{18}$$

The overall DISCO$_m$ estimator is the average of $m \leq n$ local correlations:

$$DISCO_m(X, Y \mid Z) = \frac{1}{m} \sum_{i=1}^m \frac{\widehat{\text{dCov}}_{DISCO}(X, Y \mid Z_i)}{\sqrt{\widehat{\text{dCov}}_{DISCO}(X, X \mid Z_i) \widehat{\text{dCov}}_{DISCO}(Y, Y \mid Z_i)}}. \tag{19}$$

**Proposition 2.** *Using the RBF kernel and standard assumptions of kernel regression, the DISCO$_m$ estimator consistently estimates $\text{dCor}^2(X, Y \mid Z)$. See Appendix E.3 for a proof.*

**Method 2: sDISCO.** Define the weighted means:

$$\bar{a}_{i.} = \frac{1}{w_{i.}} \sum_k^n w_{ik} a_{ik}, \quad \bar{a}_{.j} = \frac{1}{w_{.j}} \sum_k^n w_{kj} a_{kj}, \quad \bar{a}_{..} = \frac{1}{S} \sum_{k,\ell}^n w_{k\ell} a_{k\ell}, \tag{20}$$

and compute the globally centered matrix:

$$A_{ij} = a_{ij} - \bar{a}_{i.} - \bar{a}_{.j} + \bar{a}_{..}, \tag{21}$$

with $B_{ij}$ defined analogously. Unlike DISCO$_m$, here we do not use a single reference $z_i$; instead, we employ the full weight matrix $w_{ij}$, thereby mixing the centerings. The global conditional squared distance covariance is then

$$\widehat{\text{dCov}}_{eff}(X, Y \mid Z) = \frac{1}{S} \sum_{i,j}^n w_{ij} A_{ij} B_{ij}, \tag{22}$$

and the corresponding conditional distance correlation is

$$sDISCO(X, Y \mid Z) = \frac{\widehat{\text{dCov}}_{eff}(X, Y \mid Z)}{\sqrt{\widehat{\text{dCov}}_{eff}(X, X \mid Z) \widehat{\text{dCov}}_{eff}(Y, Y \mid Z)}}. \tag{23}$$

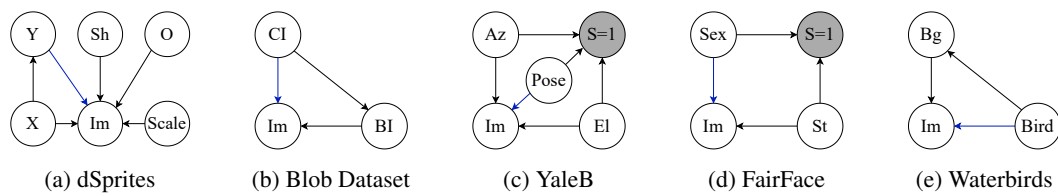

(a) dSprites      (b) Blob Dataset      (c) YaleB      (d) FairFace      (e) Waterbirds

Figure 3: Causal graphs used in our experiments across different datasets. Blue path means causally task-relevant, going from target to our input (image).

**Proposition 3.** *Using the RBF kernel under the standard assumptions of kernel regression, the sDISCO estimator consistently estimates* $\mathrm{dCor}^2(X, Y \mid Z)$. *See Appendix E.4 for an intuitive proof sketch.*

In summary, both $\mathrm{DISCO}_m$ and sDISCO provide reliable estimators of conditional dependence for random vectors of arbitrary dimensionality. Under appropriate conditions, a value of zero corresponds exactly to conditional independence.

### 3.3 PRACTICAL APPLICATION FOR CAUSAL STABILITY

Since solving the constrained optimization in Eq. 8 directly is infeasible, we adopt a regularization approach. Specifically, we add $\mathrm{DISCO}_m$ or sDISCO as a penalty term, thereby minimizing the prediction loss and conditional dependence jointly. With variables from SAM, namely $Y$ (target), $\mathbf{B}$ (bias/shortcuts), and $\hat{Y} = g_\theta(X)$ (model outputs), we optimize

$$\min_\theta \sum L\left(Y, \hat{Y}\right) + \lambda\, DISCO_m(\hat{Y}, \mathbf{B} \mid Y), \tag{24}$$

with an analogous formulation for sDISCO.

This framework is applicable to any black-box, gradient-based training procedure such as deep neural networks. Both $\mathrm{DISCO}_m$ and sDISCO introduce a hyperparameter $\lambda$ that controls the regularization strength. In addition, both methods require a kernel parameter $\sigma_Y$ over the conditioning variable $Y$. The $\mathrm{DISCO}_m$ variant further introduces a parameter $m \leq n$, which specifies the number of local measures averaged to form the estimator. For all the experiments, we fix $m$ to be 20 percent of the batch-size for the task at hand.

## 4 EXPERIMENTS

In this section, we evaluate the efficacy of $\mathrm{DISCO}_m$ and sDISCO for bias mitigation. We design experiments on six datasets that vary in realism, bias sources, and the non-linear relationships between target and bias attributes. This creates a diverse and challenging testbed for assessing causal stability. We benchmark against seven representative baselines, detailed below, and additionally leverage SAM for pathway-specific counterfactual analysis in fully controlled simulation settings. This allows us to assess not only empirical performance but also the causal properties of our methods in the presence of known and unknown biases.

### 4.1 DATASETS

Figure 3 illustrates the causal structures underlying our datasets. Across all datasets, except MNLI, we follow the standard evaluation protocol in domain generalization and bias mitigation (Sagawa et al., 2019; Arjovsky et al., 2019; Ganin et al., 2016): we train on a biased dataset, select models on a domain-shifted unbiased validation set, and report final results on an unseen unbiased test set. For these datasets, we always add label noise to make the task harder and to make models to latch on spurious correlations more easily. The full details for our MNLI dataset can be found in the Appendix F.1.6 as this dataset deviates from the rest in important aspects.

**dSprites.** Based on (Matthey et al., 2017), we construct images of geometric shapes (Sh) with different orientation (O), scale (Sc), y-position (Y), and x-position (X). The regression target, Y,

nonlinearly determines the object's y-position while X acts as a confounder. The SCM is:

$$U_x \sim \text{Uniform}\left(0, \tfrac{\pi}{2}\right), \quad U_y \sim \mathcal{N}\left(0, 0.15^2\right), \quad U_{sc} \sim \text{Uniform}(0.5, 0.7),$$

$$U_\theta \sim \text{Uniform}(0, 360°), \quad U_{\text{shape}} \sim \text{UniformDiscrete}\{\text{square}, \text{ellipse}, \text{heart}\},$$

$$\varepsilon_x \sim \mathcal{N}\left(0, 0.01^2\right), \quad \varepsilon_{y1} \sim \mathcal{N}\left(0, 0.1^2\right), \quad \varepsilon_{y2} \sim \mathcal{N}\left(0, 0.2^2\right)$$

$$x = \sin(U_x), \quad y = x^2 + U_y, \quad \text{Im} := f\left(x + \varepsilon_x, \ \exp(y + \varepsilon_{y1}) + \varepsilon_{y2}, U_{scale}, U_{shape}, U_\theta\right).$$

**Blob dataset.** Inspired by Adeli et al. (2021), we generate synthetic images with two Gaussian blobs. One blob is causally related to the regression target, causal intensity (CI), while the other acts as a spurious mediator, bias intensity (BI). The equations of the simulator are:

$$U_{\text{causal}} \sim \text{Unif}(0, 1), \quad U_{\text{bias}} \sim \mathcal{N}(0, 0.1^2), \quad \varepsilon_{\text{causal}} \sim \mathcal{N}(0, 0.1^2)$$

$$CI := U_{\text{causal}}, \quad BI := CI + U_{\text{bias}}, \quad Im := f(\exp(V_{\text{causal}} + \varepsilon_{\text{causal}}), \exp(V_{\text{bias}})).$$

**YaleB.** From the extended YaleB dataset (Georghiades et al., 2001; Yale, 2001), we predict face pose (collapsed into three categories: frontal, slightly left, maximally left). Azimuth (Az) and elevation (El) of the light source serve as continuous bias variables. We introduce a collider through a sampling mechanism:

$$\text{azimuth, elevation, pose} \sim \mathcal{D}_{\text{dataset}}, \quad X = (\text{azimuth}, \text{elevation}), \quad s = v^\top \text{Standardize}(X),$$

$$z = \text{QuantilePartition}(s, 3) \in \{0, 1, 2\}, \quad \Pr(S = 1 \mid z, y) = \begin{cases} 1.0 & \text{if } y = z, \\ 0.05 & \text{if } y \neq z. \end{cases}$$

**FairFace.** We use FairFace (Karkkainen & Joo, 2021) to predict sex as the target. Skin tone (light vs. dark) acts as a bias through selection bias. The collider is defined by:

$$P(S = 1 \mid Y, B) = \begin{cases} 0.9 & \text{if } Y = B \text{ (aligned)}, \\ 0.1 & \text{if } Y \neq B \text{ (misaligned)}. \end{cases}$$

**Waterbirds.** We adopt the Waterbirds dataset (Sagawa et al., 2019), constructed by overlaying birds (Welinder et al., 2010; Wah et al., 2011) on land or water backgrounds (Zhou et al., 2017). The background (Bg) acts as a collider bias:

$$bird \sim \text{Bernoulli}(0.5), \quad background \mid bird \sim \text{Bernoulli}\left(0.9 \cdot bird + 0.1 \cdot (1 - bird)\right).$$

**MNLI.** Finally, we also include the MNLI dataset (Williams et al., 2018). In this dataset, the task is to predict entailment of a sentence given another one. The bias in this task are negation keywords, that are highly correlated with entailment. We closely follow Sagawa et al. (2019) in designing our experiments. Full details for this setup are given in the Appendix F.1.6.

## 4.2 BIAS MITIGATION BASELINES

| | Bias Attribute | | | Target Attribute | | | Attribute Setting | |
|---|---|---|---|---|---|---|---|---|
| **Method** | Bin | Cat | Cont | Bin | Cat | Cont | Single | Multi |
| GDRO, Fishr, IRM | ✓ | ✓ | ✗ | ✓ | ✓ | ✗ | ✓ | ✓ |
| C-MMD | ✓ | ∼ | ✗ | ✓ | ∼ | ✗ | ✓ | ∼ |
| Adversarial | ✓ | ✓ | ✓ | ✓ | ✓ | ✓ | ✓ | ∼ |
| DISCO (both), HSCIC, CIRCE | ✓ | ✓ | ✓ | ✓ | ✓ | ✓ | ✓ | ✓ |

Table 1: Comparison of bias mitigation methods across bias/target types (binary, categorical, continuous) and attribute settings. ✓ = supported, ✗ = not supported, ∼ = works, but with reduced efficiency.

We benchmark against seven baselines: adversarial learning (Ganin et al., 2016; Wang et al., 2019; Adeli et al., 2021), GDRO (Sagawa et al., 2019), and three recent dependence-penalization methods: HSCIC (Quinzan et al., 2022), CIRCE (Pogodin et al., 2022), and c-MMD (Kaur et al., 2022; Makar

& D'Amour, 2022; Veitch et al., 2021). We further adapt two models from the domain generalization field, namely Fishr (Rame et al., 2022) and IRM (Arjovsky et al., 2019), to operate on groups, exactly as GDRO does. All methods share the same backbone (ResNet-18, pretrained on ImageNet (He et al., 2016; Deng et al., 2009) for real images, untrained for synthetic datasets). For the Blob dataset, we use a smaller ResNet adapted to low resolution. For MNLI, we use the pretrained TinyBERT from Jiao et al. (2020) as the backbone.

## 4.3 RESULTS

| Model | dSprites ($R^2$) | Blob ($R^2$) | YaleB (BAcc) | FairFace (BAcc) | Waterbirds (BAcc) |
|---|---|---|---|---|---|
| ResNet (Bias) | $0.417 \pm 0.037$ | $0.281 \pm 0.017$ | $0.607 \pm 0.012$ | $0.744 \pm 0.010$ | $0.730 \pm 0.032$ |
| ResNet (No Bias) | $0.757 \pm 0.023$ | $0.889 \pm 0.001$ | $0.976 \pm 0.009$ | $0.904 \pm 0.007$ | $0.908 \pm 0.008$ |
| $\text{DISCO}_m$ | $\mathbf{0.688 \pm 0.018}$ | $\underline{0.759 \pm 0.015}$ | $\mathbf{0.804 \pm 0.027}$ | $\mathbf{0.860 \pm 0.006}$ | $\underline{0.867 \pm 0.009}$ |
| sDISCO | $\underline{0.619 \pm 0.011}$ | $\mathbf{0.788 \pm 0.017}$ | $\underline{0.694 \pm 0.020}$ | $\underline{0.854 \pm 0.005}$ | $0.863 \pm 0.006$ |
| Adversarial | $0.467 \pm 0.034$ | $0.647 \pm 0.046$ | $0.692 \pm 0.011$ | $0.846 \pm 0.006$ | $0.845 \pm 0.017$ |
| GDRO | – | – | – | $0.840 \pm 0.012$ | $0.865 \pm 0.009$ |
| c-MMD | – | – | – | $0.824 \pm 0.006$ | $0.853 \pm 0.019$ |
| CIRCE | $0.606 \pm 0.021$ | $0.748 \pm 0.015$ | $0.681 \pm 0.026$ | $0.822 \pm 0.006$ | $\mathbf{0.894 \pm 0.010}$ |
| HSCIC | $0.568 \pm 0.033$ | $0.470 \pm 0.022$ | $0.708 \pm 0.035$ | $0.748 \pm 0.013$ | $0.853 \pm 0.012$ |
| IRM | – | – | – | $0.821 \pm 0.017$ | $0.858 \pm 0.009$ |
| Fishr | – | – | – | $0.727 \pm 0.004$ | $0.814 \pm 0.012$ |

Table 2: Performance of all models across 5 datasets. Rows 1–2 (shaded) show naive baselines: *Backbone (Biased)* = lower bound, *Backbone (Unbiased)* = upper bound. Regression datasets are reported with $R^2$, classification with Balanced Accuracy. Values are mean $\pm$ standard deviation across runs. Best results per dataset are bolded.

Tab. 2 shows that $\text{DISCO}_m$ and sDISCO consistently outperform adversarial and dependence-penalization baselines, and are competitive with or superior to GDRO and CIRCE. $\text{DISCO}_m$ achieves the best results on dSprites and FairFace, while sDISCO dominates on Blob. CIRCE is strong on Waterbirds, which might be explained by the higher hyperparameter search budget we assigned to it, as elaborated below. HSCIC and adversarial training generally lag behind. The full results for MNLI is presented in the Appendix F.1.6 due to page limits. For MNLI, again, we see that the DISCO variants are again very competitive and secure the second spot, while only GDRO has a slight advantage of 0.4% on worst group accuracy.

Besides pure debiasing strength, an additional practical advantage of $\text{DISCO}_m$ and sDISCO is their relatively simple hyperparameterization. For all models except HSCIC and CIRCE, we conducted hyperparameter searches with a maximum budget of 36 runs. This budget is already larger than what is commonly used in prior work on bias mitigation baselines (Sagawa et al., 2019; Kaur et al., 2022; Makar et al., 2022), ensuring a fair and thorough comparison. By contrast, both HSCIC and CIRCE have a substantially larger number of hyperparameters, leading to a rapidly expanding search space when performing grid search. To account for this, we allocated them a larger budget of 100 runs, giving these methods an advantage when it comes to hyperparameter search budget. Despite this additional budget, $\text{DISCO}_m$ and sDISCO still achieve consistently superior or competitive performance, highlighting their robustness and efficiency. Details about the experiments and hyperparameters can be found in the appendix F.2.

## 4.4 PATH ANALYSIS AND UNOBSERVED BIAS

While classical bias mitigation experiment results are reported on an unbiased test set, one can also perform pathway analysis if we have access to the data-generating process. These pathway-specific effects allow us more in-depth insights into the causal stability of the tested methods. To illustrate the analytic power of SAM, we conduct pathway-specific counterfactual analysis in a controlled setting on the dSprites dataset, see (Figure 4b), as we have full control over the dataset, including the generation of true counterfactuals. We compare a naive ResNet to two variants of sDISCO: one aware of both bias variables ($\text{sDISCO}_{XY}$) and one aware of only one bias ($\text{sDISCO}_X$), thus the latter

| Model | $S_X$ | $S_Y$ | $Acc_{ctf}$ |
|---|---|---|---|
| sDISCO$_{X,Y}$ | 0.066 | 0.067 | 0.98 |
| sDISCO$_X$ | 0.117 | 0.154 | 0.83 |
| ResNet | 0.302 | 0.275 | 0.65 |

(a) Counterfactual sensitivity measures.

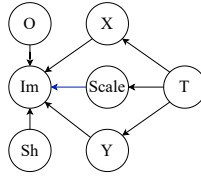

(b) Multi Bias Scenario dSprites.

Figure 4: Comparison of sensitivity measures and the bias scenario.

model will be simulated in an unobserved confounder setting. We visualized some counterfactuals in Fig. F.8. We predict $Scale$ in this setting after we binarize it.

Formally, we quantify model sensitivity to counterfactual changes of a bias variable $X$ via the following measure:

$$S_X(\theta) := \mathbb{E}_{u\sim U}\, \mathbb{E}_{x\sim \mathrm{Unif}(\mathrm{supp}(X))} \left[ \left| P(\hat{Scale}(u); \theta_m) - P(\hat{Scale}_x(u); \theta_m) \right| \right], \qquad (25)$$

and analogously define $S_Y(\theta)$ for the variable $Y$. Intuitively, $S_X$ and $S_Y$ measure the average discrepancy in predictions under counterfactual interventions on $X$ or $Y$, respectively. In addition, we report counterfactual accuracy,

$$Acc_{ctf} := \mathbb{E}_{u\sim U}\, \mathbb{E}_{s\sim \mathrm{Unif}(\mathrm{supp}(Scale))} \left[ \mathbf{1}\{s = \hat{Scale}_s(u)\} \right], \qquad (26)$$

which evaluates whether predictions remain consistent with ground-truth outcomes under counterfactual changes of the causally relevant target variable $Scale$.

Note that out methods still work solely on observational data. But with the help of this controlled experiment on dSprites, we can even simulate true counterfactuals, and show, that our proposed method actually provides causal stability in terms of counterfactuals.

Counterfactual sensitivity (Tab. 4a) reveals that ResNet predictions are highly sensitive to both spurious paths, while sDISCO$_{XY}$ nearly eliminates such effects and achieves high counterfactual accuracy. Even sDISCO$_X$, which only observes one of the two biases, substantially reduces sensitivity on both paths, though residual effects remain. This aligns with our theoretical results: unknown biases cannot be removed, but blocking all known biases is the best we can and should do. In the appendix F.4, we further highlight how this pathway analysis can be used to understand why some models failed in learning causally stable relationships in detail.

## 5 CONCLUSION

We introduced the Standard Anti-Causal Model (SAM), a unifying framework to analyze bias mechanisms in prediction tasks. From SAM, we derived a conditional independence criterion that guarantees *causal stability*, ensuring predictions rely solely on stable direct effects.

To optimize towards causal stability, we proposed two new estimators of conditional distance correlation: DISCO$_m$ and sDISCO. While distance correlation had previously been computationally impractical, our methods make it efficient, differentiable, and scalable to deep learning. DISCO$_m$ provides accurate estimation, while sDISCO offers a single-shot, highly efficient alternative.

Across six datasets with diverse bias structures, DISCO$_m$ and sDISCO consistently outperform or are competitive against state-of-the-art baselines, while requiring fewer hyperparameters and supporting arbitrary target–bias types. Finally, SAM enables pathway-specific counterfactual analysis, providing deeper insight into model behavior under interventions.

## 6 REPRODUCIBILITY STATEMENT

All experimental settings and implementation details are described in Section 4, with further clarifications provided in the appendix. To facilitate verification, we have uploaded the complete

codebase as part of the supplementary material for this submission, and we will release it publicly at `https://github.com/***` upon acceptance. The codebase contains all configuration files and scripts required to exactly reproduce the reported results in a fully automated manner including data generation and sampling procedures. Our experiments were implemented using PyTorch[2], PyTorch Lightning[3], and Hydra[4]. The mathematical proofs can be found in our appendix, as referenced in the main text.

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

APPENDIX

# A  USE OF LLMS

We used LLMs for this manuscript to polish our writing on a sentence and paragraph level. It was used to intelligently find and eliminate spelling mistakes, improve sentence structure, and to refine the flow of text when necessary.

The used tools are: Grammarly (`https://www.grammarly.com/`) and ChatGPT from OpenAI (`https://chatgpt.com/`).

# B  CONTEXTUALIZATION AND RELATED WORK

We propose both an analytical causal framework to understand anti-causal prediction from a path-way analysis point of view, and also provide a solution to estimate bias free predictors, as well.

One can thus interpret our work as an candidate of causal representation learning (CRL) methods, or by a causally motivated shortcut/bias mitigation method.

## B.1  SAM AND DISCO IN CONTEXT OF CRL

Our work aligns with the central objective of Causal Representation Learning (CRL), which is to learn representations that are robust and generalize across environments by leveraging causal principles. While CRL is a broad field, distinct sub-goals exist, such as: (a) *Generative and Disentangled CRL*, which aims to learn the full causal generative process or disentangle independent causal mechanisms (Sanchez & Tsaftaris, 2022; Komanduri et al., 2023); (b) *Latent Confounder CRL*, which focuses on inferring causal effects when key confounders are unobserved (Wang et al., 2024; Louizos et al., 2017; Kompa et al., 2022); and (c) *Causal Discovery*, which attempts to infer the graph structure from high-dimensional data (Lagemann et al., 2023). These are just very limited sub-fields, we note that there are many more highly relevant CRL subfields (Schölkopf et al., 2021).

Our framework occupies a specific and distinct niche: **Discriminative CRL with observed bias**. Unlike generative approaches, we do not attempt to model the full data-generating process or reconstruct the input. Instead, we demonstrate that for the specific problem of robust prediction, one can avoid the complexity of full generative modeling by adhering to a three-step pipeline:

1. **Modeling:** We propose the Standard Anti-Causal Model (SAM) to formally characterize the data-generating process and its biases $B$.

2. **Criterion:** From SAM, we derive a formal observational criterion, *Causal Stability* ($\hat{Y} \perp B \mid Y$), which we prove is sufficient to ensure that counterfactual indirect and spurious effects vanish, isolating the stable direct effect.

3. **Estimation:** We develop DISCO as an efficient regularization technique to optimize deep models toward this causal criterion.

A key distinction of our approach is the assumption that the bias $B$ is observed. While some CRL methods operate under latent confounding, the assumption of known attributes is foundational to the specific subfield of bias mitigation and is highly realistic in many high-stakes applications. These include fairness auditing (where sensitive attributes like age or sex are collected for compliance), scientific and industrial settings (where metadata regarding sensors or collection times is available), and standard benchmarks explicitly designed to model known spurious correlations (e.g., Waterbirds, FairFace).

## B.2  SAM AND DISCO IN CONTEXT OF SHORTCUT MITIGATION

### B.2.1  SAM

Our work provides a structural foundation that generalizes and clarifies findings from prior influential works in shortcut mitigation, specifically those of Makar et al. (2022); Makar & D'Amour (2022); Veitch et al. (2021); Puli et al. (2021).

**Comparison to Makar et al. (2022); Makar & D'Amour (2022):** Makar & D'Amour (2022) connect risk invariance (a robustness criterion) with separation ($f(X) \perp B \mid Y$) (a fairness criterion). Their analysis operates primarily at the *distributional* level (associated with Rung 1 of Pearl's Ladder of Causation), describing *what* statistical property a robust model should possess. In contrast, SAM operates at the *structural and counterfactual* level (Rung 2 and 3). By making our assumptions explicit via a Structural Causal Model, we analyze the flow of information along specific causal pathways. Since structural guarantees imply distributional ones, our work provides the deeper causal mechanism for *why* the separation principle works: blocking specific spurious paths in the graph necessitates the resulting distributional independence.

**Comparison to Veitch et al. (2021):** Veitch et al. (2021) formalize stress tests using a notion of *unit-level counterfactual invariance*, requiring $f(X(z)) = f(X(z'))$. While theoretically robust, this criterion relies on unobserved potential outcomes and is often impractical to enforce directly. Veitch et al. (2021) show that observational conditional independence is a *necessary* but not *sufficient* signature for this invariance. Our work bridges this gap by defining a more practical notion of invariance tied to observable attributes and path-specific mechanisms within SAM. In our setting, we prove that the conditional independence criterion is *sufficient* to block the transmission of spurious information through specific causal pathways (Theorem 1), making the guarantee attainable for practitioners.

**Comparison to Puli et al. (2021):** Puli et al. (2021) (NURD) argue that enforcing conditional independence is too restrictive. They provide examples where an optimal representation $r(x)$ must depend on the nuisance $z$ to maximize mutual information with the label. However, this critique applies to *generative* or *representational* goals, where the aim is to retain as much information about the input $X$ as possible.

1. **Discriminative vs. Generative Targets:** DISCO is designed for a discriminative task. We apply the independence constraint to the *final prediction* $\hat{Y}$, not to an intermediate representation $r(x)$. We allow the model's internal layers to utilize bias information if necessary, provided the final output is purged of spurious signal.

2. **Stability vs. Information:** In the example provided by Puli et al. (2021), the "optimal" representation depends on the nuisance. Consequently, a predictor built on such a representation would also depend on the nuisance. By definition, this results in a predictor that is not causally stable against shifts in that nuisance.

Our objective (maximizing the stable direct effect, $ctf\text{-}DE$) explicitly accepts the trade-off that we do not wish to model the full input $X$. Instead, we isolate the path $Y \to X \to \hat{Y}$ while forcing counterfactual indirect ($ctf\text{-}IE$) and spurious ($ctf\text{-}SE$) effects to zero. This ensures that the model utilizes the bias $B$ only insofar as it helps extract the stable direct signal, providing a guarantee that is aligned with robust discriminative prediction rather than generative reconstruction.

### B.2.2 DISCO

Finally, within the realm of bias mitigation, we want to further highlight that the causal analysis is not our only contribution. We additionally propose the efficient conditional independence penalty via conditional distance correlation. While we compare against classical baselines such as GDRO (Sagawa et al., 2019), IRM (Arjovsky et al., 2019), Fishr (Rame et al., 2022), and adversarial bias mitigation (Ganin et al., 2016; Wang et al., 2019; Adeli et al., 2021), we especially include the modern approaches based on Reproducing Kernel Hilbert Spaces (RKHs) that also penalize conditional dependence directly. In this category we compare against conditional MMD, called C-MMD (Kaur et al., 2022; Makar & D'Amour, 2022; Veitch et al., 2021), conditional Hilbert-Schmidt Independence Criterion (HSCIC) (Quinzan et al., 2022), and CIRCE (Pogodin et al., 2022) (an efficient variant of HSCIC).

We especially want to highlight that our eDISCO and DISCO$_m$ estimators, together with adversarial bias mitigation, HSCIC and CIRCE, are extremely flexible, as they can be applied to any types of targets and bias attributes, while all the other methods, including the classical gold-standard baselines, always require categorical attributes.

Our methods have significantly less hyperparameters than CIRCE and HSCIC, while being more flexible than the classical methods and consistently providing the best or second best results in our 6 different datasets with different data, bias, and target types.

## C  CAUSAL GRAPH AND ASSUMPTIONS

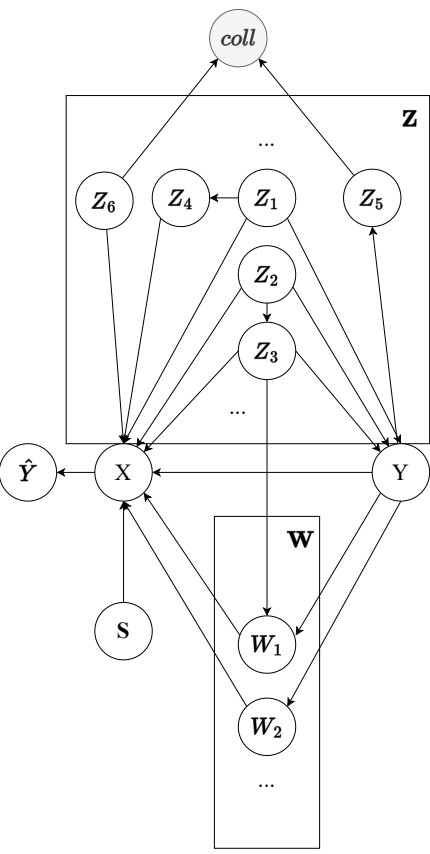

Figure C.1: The main setting we consider in anti-causal prediction. Y is the target we want to predict using X only. **W** is a set of variables that only permits mediated effects from Y to X. **Z** contains nodes that lie on open paths between Y and X but are non-directed, i.e. any kind of nodes that lie on open back-door paths between $X$ and $Y$, or any kinds of paths that are opened due to conditioned colliders. Nodes in **Z** are allowed to have arbitrary connections in between them, as long as **Z** stays a valid adjustment set for (Y,X), and the graph is still a directed acyclic graph (DAG). Nodes in **W** are mediators. Nodes in **S** are variables that are d-separation irrelevant between Y and X (and Y and $\hat{Y}$ respectively). We can always ignore the variables in **S** for our settings as they do not harm or inform anything for our prediction setup. Thus, these variables are omitted in our analysis and discussions in the main manuscript.

### C.1  VALID ADJUSTMENT SET

**Characterization of Valid Adjustment Sets (Shpitser et al., 2012)** Let $Z$ be a set of nodes in a causal graph. Then, $Z$ is a valid adjustment set for $(Y, X)$ (implying $Y \cap X = \text{pa}_Y \cap X = \emptyset$) if and only if it satisfies the following conditions:

  (i) $Z$ contains no node $R \notin Y$ on a proper causal path from $Y$ to $X$ nor any of its descendants in $G_Y$,

  (ii) $Z$ blocks all non-directed paths from $Y$ to $X$.

## D    CAUSAL INVARIANCE AS CONDITIONAL INDEPENDENCE

In the following, we will prove the propositions in Sec.2.

### D.1    TOTAL VARIANCE DECOMPOSITION

**Proposition 1 (TV-Decomposition)** The TV can be decomposed into direct, indirect, and spurious components by

$$TV_{y_0,y_1}(\hat{y}) = ctf\text{-}DE_{y_0,y_1}(\hat{y}|y) - ctf\text{-}IE_{y_1,y_0}(\hat{y}|y) - ctf\text{-}SE_{y_1,y_0}(\hat{y}). \tag{27}$$

*Proof.* We first show that $ctf\text{-}TE_{y_0,y_1}(\hat{y}|y_0)$ can be split into its direct and indirect components as given in definition 2.1.

$$ctf\text{-}TE_{y_0,y_1}(\hat{y}|y_0) = P(\hat{y}_{y_1}|y_0) - P(\hat{y}_{y_0}|y_0) \tag{28}$$
$$= P(\hat{y}_{y_1}|y_0) - P(\hat{y}_{y_1,w_{y_0}}|y_0) + P(\hat{y}_{y_1,w_{y_0}}|y_0) - P(\hat{y}_{y_0}|y_0) \tag{29}$$
$$= ctf\text{-}DE_{y_0,y_1}(\hat{y}|y_0) - ctf\text{-}IE_{y_1,y_0}(\hat{y}|y_0). \tag{30}$$

We now can finally show that the total variance can be split into total effects and spurious effects, as given in the following:

$$TV_{y_0,y_1}(\hat{y}) = P(\hat{y}|y_1) - P(\hat{y}|y_0) \tag{31}$$
$$= P(\hat{y}|y_1) - P(\hat{y}_{y_1}|y_0) + P(\hat{y}_{y_1}|y_0) - P(\hat{y}|y_0) \tag{32}$$
$$= P(\hat{y}_{y_1}|y_0) - P(\hat{y}_{y_0}|y_0) + P(\hat{y}_{y_1}|y_1) - P(\hat{y}_{y_1}|y_0) \tag{33}$$
$$= ctf\text{-}TE_{y_0,y_1}(\hat{y}|y_0) - ctf\text{-}SE_{y_1,y_0}(\hat{y}). \tag{34}$$

In total, we arrive at

$$TV_{y_0,y_1}(\hat{y}) = ctf\text{-}DE_{y_0,y_1}(\hat{y}|y_0) - ctf\text{-}IE_{y_1,y_0}(\hat{y}|y_0) - ctf\text{-}SE_{y_1,y_0}(\hat{y}). \tag{35}$$

$\square$

**Proposition 4.** *($ctf\text{-}IE_{y_1,y_0}(\hat{y}|y_0,\mathbf{w},\mathbf{z})$ is stronger than $ctf\text{-}IE_{y_1,y_0}(\hat{y}|y_0)$) The expression $ctf\text{-}IE_{y_1,y_0}(\hat{y}|y_0,\mathbf{w},\mathbf{z})$ is a more fine-grained measure of indirect effect than $ctf\text{-}IE_{y_1,y_0}(\hat{y}|y_0)$, as the former captures more nuances of effects than the latter one. Whenever $ctf\text{-}IE_{y_1,y_0}(\hat{y}|y_0,\mathbf{w},\mathbf{z})$ is zero, $ctf\text{-}IE_{y_1,y_0}(\hat{y}|y_0)$ is zero. The converse is not true.*

*Proof.* We can further extend the total variation formula by the total probability theorem, and we arrive at

$$TV_{y_0,y_1}(\hat{y}) = \sum_{\mathbf{w},\mathbf{z}} ctf\text{-}DE_{y_0,y_1}(\hat{y}|y_0,\mathbf{w},\mathbf{z})P(\mathbf{w},\mathbf{z}|y_0) \tag{36}$$

$$- \sum_{\mathbf{w},\mathbf{z}} ctf\text{-}IE_{y_1,y_0}(\hat{y}|y_0,\mathbf{w},\mathbf{z})P(\mathbf{w},\mathbf{z}|y_0) \tag{37}$$

$$- ctf\text{-}SE_{y_1,y_0}(\hat{y}) \tag{38}$$

It is therefore easy to see that $ctf\text{-}IE_{y_1,y_0}(\hat{y}|y_0,\mathbf{w},\mathbf{z}) = 0 \implies ctf\text{-}IE_{y_1,y_0}(\hat{y}|y_0) = 0$, but there converse is not true in general. Proof of the falsity of the converse can be given by simple counterexamples. An interested reader can find some in (Plecko & Bareinboim, 2022). $\square$

### D.2    JUSTIFICATION CONDITIONAL INDEPENDENCE AS CONSTRAINT

**Proposition (Conditional Independence for unbiased Predictors)** SAM is given. Assume we have a prediction model $g_\theta$ that maps from the input $X$ to our target. Let's treat the predictions of our model as a random variable called $\hat{Y}$. If the prediction satisfies $\hat{Y} \perp \mathbf{W}, \mathbf{Z} \mid Y$, then $ctf\text{-}IE_{y_1,y_0}(\hat{y} \mid y) = ctf\text{-}SE_{y_1,y_0}(\hat{y}) = 0$, for any $y, y_0, y_1, \hat{y}$ in their ranges. The conditional independence criterion even ensures a stricter criterion, namely, $\hat{Y} \perp \mathbf{W}, \mathbf{Z} \mid Y \implies ctf\text{-}IE_{y_1,y_0}(\hat{y} \mid y, \mathbf{z}, \mathbf{w}) = 0$, for any $y, y_0, y_1, \hat{y}, \mathbf{w}, \mathbf{z}$ in the ranges of their respective random variables.

Outline

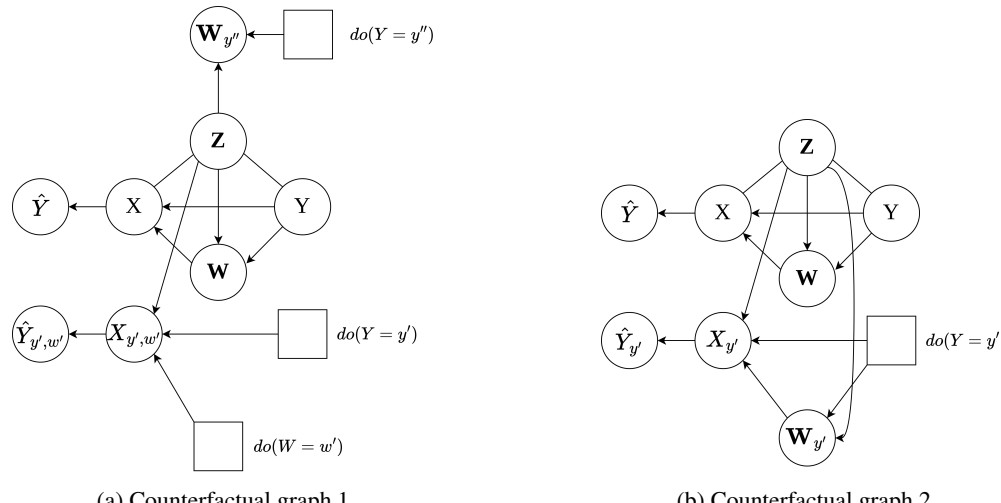

(a) Counterfactual graph 1          (b) Counterfactual graph 2

Figure D.2: Counterfactual Graphs needed for the proofs of propositions 2 and 3.

- Prop. 4 tells us that $ctf\text{-}IE_{y_1,y_0}(\hat{y}|y_0, \mathbf{w}, \mathbf{z})$ is stronger than $ctf\text{-}IE_{y_1,y_0}(\hat{y}|y_0)$. Therefore, we will show the sufficiency of the stronger case, which will automatically apply to the weaker one.

- We need to derive some (in)dependence criteria based on the graph. We will use the theory from twin graphs (parallel worlds graphs) (Balke & Pearl, 1994; Avin et al., 2005) and their extensions made in the make-cg algorithm (Shpitser & Pearl, 2012).

- Another useful tool from (Correa et al., 2021) will be used. We will refer to their Theorem 1 as counterfactual unnesting (*ctf-unnesting*) when we use it.

- We first show $\hat{Y} \perp \mathbf{W}, \mathbf{Z}|Y \implies ctf\text{-}IE_{y_1,y_0}(\hat{y}|y, \mathbf{z}, \mathbf{w}) = 0$

- We afterwards show $\hat{Y} \perp \mathbf{W}, \mathbf{Z}|Y \implies ctf\text{-}SE_{y_1,y_0}(\hat{y}) = 0$

We begin with the graphical independence criteria. Relevant interventions and observations are in this case $\gamma = (\hat{Y}_{y',\mathbf{w}'}, \mathbf{w}_{y''}, y, \mathbf{w}, \mathbf{z})$. From the graph in Fig. D.2, we can conclude that $\hat{Y}_{y',\mathbf{w}'} \perp \mathbf{W}_{y''}, \mathbf{W}, Y \mid \mathbf{Z}$. Having these relations, we can prove that $\hat{Y} \perp \mathbf{W}, \mathbf{Z}|Y \implies ctf\text{-}IE_{y_1,y_0}(\hat{y}|y, \mathbf{z}, \mathbf{w}) = 0$.

*Proof.*

$$ctf\text{-}IE_{y_1,y_0}(\hat{y}|y,\mathbf{w},\mathbf{z}) = P(\hat{y}_{y_1,\mathbf{w}_{y_0}}|y,\mathbf{z},\mathbf{w}) - P(\hat{y}_{y_1}|y,\mathbf{z},\mathbf{w})$$

$$= P(\hat{y}_{y_1,\mathbf{w}_{y_0}}|y,\mathbf{z},\mathbf{w}) - P(\hat{y}_{y_1,\mathbf{w}_{y_1}}|y,\mathbf{z},\mathbf{w})$$

$$= \sum_{\mathbf{w}'} P(\hat{y}_{y_1,\mathbf{w}'}, \mathbf{W}_{y_0} = \mathbf{w}'|y,\mathbf{z},\mathbf{w})$$

$$\qquad - \sum_{\mathbf{w}''} P(\hat{y}_{y_1,\mathbf{w}''}, \mathbf{W}_{y_1} = \mathbf{w}''|y,\mathbf{z},\mathbf{w}) \qquad\qquad \textit{ctf-unnesting}$$

$$= \sum_{\mathbf{w}'} P(\hat{y}_{y_1,\mathbf{w}'}|y,\mathbf{z},\mathbf{w})P(\mathbf{W}_{y_0} = \mathbf{w}'|y,\mathbf{z},\mathbf{w})$$

$$\qquad - \sum_{\mathbf{w}''} P(\hat{y}_{y_1,\mathbf{w}''}|y,\mathbf{z},\mathbf{w})P(\mathbf{W}_{y_1} = \mathbf{w}''|y,\mathbf{z},\mathbf{w}) \quad \hat{Y}_{y',\mathbf{w}'} \perp \mathbf{W}_{y''} \mid \mathbf{Z}$$

$$= \sum_{\mathbf{w}'} P(\hat{y}_{y_1,\mathbf{w}'}|y_1,\mathbf{z},\mathbf{w}')P(\mathbf{w}'|y_0,\mathbf{z}) \qquad\qquad \hat{Y}_{y',\mathbf{w}'} \perp Y, \mathbf{W} \mid \mathbf{Z},$$

$$\qquad - \sum_{\mathbf{w}''} P(\hat{y}_{y_1,\mathbf{w}''}|y_1,\mathbf{z},\mathbf{w}'')P(\mathbf{w}''|y_1,\mathbf{z}) \qquad \mathbf{W}_{y''} \perp Y, \mathbf{W} \mid \mathbf{Z}$$

$$= \sum_{\mathbf{w}'} P(\hat{y}|y_1,\mathbf{z},\mathbf{w}')P(\mathbf{w}'|y_0,\mathbf{z})$$

$$\qquad - \sum_{\mathbf{w}''} P(\hat{y}|y_1,\mathbf{z},\mathbf{w}'')P(\mathbf{w}''|y_1,\mathbf{z})$$

$$= P(\hat{y}|y_1) \sum_{\mathbf{w}'} P(\mathbf{w}'|y_0,\mathbf{z}) \qquad\qquad\qquad\qquad \hat{Y} \perp \mathbf{W}, \mathbf{Z}|Y$$

$$\qquad - P(\hat{y}|y_1) \sum_{\mathbf{w}''} P(\mathbf{w}''|y_1,\mathbf{z})$$

$$= 0$$

□

Next, we prove that $\hat{Y} \perp \mathbf{W}, \mathbf{Z}|Y \implies ctf\text{-}SE_{y_1,y_0}(\hat{y}) = 0$.

*Proof.*

$$SE_{y_1,y_0}(\hat{y}) = P(\hat{y}_{y_1} \mid y_0) - P(\hat{y}_{y_1} \mid y_1)$$

$$= \sum_{\mathbf{z}} [P(\hat{y}_{y_1} \mid y_0,\mathbf{z})P(\mathbf{z} \mid y_0) - P(\hat{y}_{y_1} \mid y_1,\mathbf{z})P(\mathbf{z} \mid y_1)] \quad \text{Law of total Prob.}$$

$$= \sum_{\mathbf{z}} [P(\hat{y} \mid y_1,\mathbf{z})P(\mathbf{z} \mid y_0) - P(\hat{y} \mid y_1,\mathbf{z})P(\mathbf{z} \mid y_1)] \qquad \hat{Y}_{y_1} \perp Y \mid \mathbf{Z}$$

$$= P(\hat{y} \mid y_1) \sum_{\mathbf{z}} P(\mathbf{z} \mid y_0) - P(\hat{y} \mid y_1) \sum_{\mathbf{z}} P(\mathbf{z} \mid y_1) \qquad \hat{Y} \perp \mathbf{W}, \mathbf{Z} \mid Y$$

$$= 0$$

□

### D.3 CONSTANT DIRECT EFFECTS GIVEN Y,Y'

**Proposition** (Constant Direct Effect for any $\mathbf{w}, \mathbf{z}$) The graph $\mathcal{G}$ and all implied assumptions are given. Assume we have a predictor $h_{\theta_h} \circ g_{\theta_g}$ and its output represented as a random variable $\hat{Y}$, such that $\hat{Y} \perp \mathbf{W}, \mathbf{Z} \mid Y$ holds. This predictor will have a constant direct effect for fixed $y_0, y_1$, for any $\mathbf{w}, \mathbf{z}$. This is compactly expressed as $ctf\text{-}DE_{y_0,y_1}(\hat{y}|y,\mathbf{w},\mathbf{z}) = ctf\text{-}DE_{y_0,y_1}(\hat{y}|y,\mathbf{w}',\mathbf{z}')$, for any $y_0, y_1, \mathbf{w}, \mathbf{z}, \mathbf{w}', \mathbf{z}'$.

*Proof.*

$$
\begin{aligned}
ctf\text{-}DE_{y_1,y_0}(\hat{y}|y,\mathbf{w},\mathbf{z}) &= P(\hat{y}_{y_1,\mathbf{w}_{y_0}}|y,\mathbf{z},\mathbf{w}) - P(\hat{y}_{y_0}|y,\mathbf{z},\mathbf{w}) \\
&= P(\hat{y}_{y_1,\mathbf{w}_{y_0}}|y,\mathbf{z},\mathbf{w}) - P(\hat{y}_{y_0,\mathbf{w}_{y_0}}|y,\mathbf{z},\mathbf{w}) \\
&= \sum_{\mathbf{w}'}[P(\hat{y}_{y_1,\mathbf{w}'},\mathbf{W}_{y_0} = \mathbf{w}'|y,\mathbf{z},\mathbf{w}) \\
&\quad - P(\hat{y}_{y_0,\mathbf{w}'},\mathbf{W}_{y_0} = \mathbf{w}'|y,\mathbf{z},\mathbf{w})] \qquad\qquad \textit{ctf-unnesting} \\
&= \sum_{\mathbf{w}'}[P(\hat{y}_{y_1,\mathbf{w}'}|y,\mathbf{z},\mathbf{w}) \qquad\qquad\qquad \mathbf{W}_{y''} \perp Y, \mathbf{W} \mid \mathbf{Z} \\
&\quad - P(\hat{y}_{y_0,\mathbf{w}'}|y,\mathbf{z},\mathbf{w})]P(\mathbf{W}_{y_0} = \mathbf{w}'|y,\mathbf{z}) \\
&= \sum_{\mathbf{w}'}[P(\hat{y}|y_1,\mathbf{z},\mathbf{w}') \qquad\qquad\qquad \hat{Y}_{y',\mathbf{w}'} \perp Y, \mathbf{W} \mid \mathbf{Z} \\
&\quad - P(\hat{y}|y_0,\mathbf{z},\mathbf{w}')]P(\mathbf{W}_{y_0} = \mathbf{w}'|y,\mathbf{z}) \\
&= [P(\hat{y}|y_1) - P(\hat{y}|y_0)]\sum_{\mathbf{w}'} P(\mathbf{W}_{y_0} = \mathbf{w}'|y,\mathbf{z}) \quad \hat{Y} \perp \mathbf{W}, \mathbf{Z} \mid Y \\
&= P(\hat{y}|y_1) - P(\hat{y}|y_0)
\end{aligned}
$$

$\square$

## E PROOFS AND SUPPLEMENTARY DEFINITIONS FOR DISCO THEORY

### E.1 SUPPLEMENTARY DEFINITIONS

**Definition 5** (Restatement of Definitions: Conditional distance covariance). *Let $P_Z$ denote the probability measure of $Z$. For $z \in \mathcal{Z}$, let $\mu_z := P_{X|Z=z}$ and $\nu_z := P_{Y|Z=z}$ be the respective conditional marginals. For $\theta_z := P_{(X,Y)|Z=z}$, define the distance centered kernels:*

$$
d_{\mu_z}(x,x') := d_{\mathcal{X}}(x,x') - a_{\mu_z}(x) - a_{\mu_z}(x') + D(\mu_z),
$$

*where $a_{\mu_z}(x) := \int d_{\mathcal{X}}(x,x')\,d\mu_z(x')$, $D(\mu_z) := \int d_{\mathcal{X}}(x,x')\,d\mu_z(x)d\mu_z(x')$, and similarly for $d_{\nu_z}$.*

*The conditional distance covariance is defined as:*

$$
\mathrm{dCov}^2(X,Y \mid Z) := \mathbb{E}_Z\left[\mathrm{dCov}^2(X,Y \mid Z = z)\right],
$$

*where for each $z \in \mathcal{Z}$,*

$$
\mathrm{dCov}^2(X,Y \mid Z = z) := \int d_{\mu_z}(x,x')\,d_{\nu_z}(y,y')\,dP_{(X,Y)|Z=z}(x,y)dP_{(X,Y)|Z=z}(x',y').
$$

**Definition 6** (Strong negative type (Lyons, 2013, §3)). *A metric space $(\mathcal{X}, d)$ has negative type if whenever two Borel probability measures $\mu_1, \mu_2$ on $\mathcal{X}$ with finite first moments satisfy*

$$
D(\mu_1 - \mu_2) := \iint d(x,x')\,d(\mu_1 - \mu_2)(x)\,d(\mu_1 - \mu_2)(x') \leq 0.
$$

*Here finite first moment means $\int d(o,x)\,d\mu_i(x) < \infty$ for some (hence any) base point $o \in \mathcal{X}$. The same metric space is of strong negative type when equality implies $\mu_1 = \mu_2$.*

**Definition 7** (Hilbert embedding (Lyons, 2013, §3)). *An isometric embedding of $(\mathcal{X}, d)$ into a real Hilbert space $\mathcal{H}$ is a map $\varphi : \mathcal{X} \to \mathcal{H}$ such that*

$$
d(x,x') = \|\varphi(x) - \varphi(x')\|_{\mathcal{H}}^2 \quad \forall x, x' \in \mathcal{X}.
$$

*By Schoenberg's theorem (Schoenberg, 1937; 1938), $(\mathcal{X}, d)$ has negative type if and only if such $\varphi$ exists.*

**Definition 8** (Tensor product space). *If $\varphi : \mathcal{X} \to \mathcal{H}_X$ and $\psi : \mathcal{Y} \to \mathcal{H}_Y$ are embeddings into real Hilbert spaces, their tensor-product embedding*

$$
\varphi \otimes \psi : \mathcal{X} \times \mathcal{Y} \to \mathcal{H}_X \otimes \mathcal{H}_Y
$$

*is defined on simple tensors by*

$$(\varphi \otimes \psi)(x,y) \;=\; \varphi(x) \,\otimes\, \psi(y),$$

*and extended linearly and by continuity. The inner product on $\mathcal{H}_X \otimes \mathcal{H}_Y$ satisfies*

$$\langle u_1 \otimes v_1,\; u_2 \otimes v_2 \rangle = \langle u_1, u_2 \rangle_{\mathcal{H}_X} \, \langle v_1, v_2 \rangle_{\mathcal{H}_Y}.$$

**Definition 9** (Barycenter map (Lyons, 2013, Prop. 3.1)). *Given an embedding $\varphi : \mathcal{X} \to \mathcal{H}$ and any signed Borel measure $\mu$ on $\mathcal{X}$ with finite first moment, define its* barycenter *(or* mean embedding*, compare to maximum mean discrepancy, HSIC, etc., see (Schrab, 2025) as*

$$\beta_\varphi(\mu) := \int_{\mathcal{X}} \varphi(x)\, \mu(dx) \;\in \mathcal{H},$$

*which is well-defined.*

E.2   PROOF OF THEOREM (CONDITIONAL INDEPENDENCE IFF ZERO COVARIANCE)

**Remark 1** (Overloaded notation for $\otimes$). *In this setting, the symbol $\otimes$ is used in two distinct contexts, following Lyons' notation (Lyons, 2013). First, for probability measures, as in $\mu_z \otimes \nu_z$, it denotes the* product measure *on the product space $\mathcal{X} \times \mathcal{Y}$. Second, for Hilbert-space-valued embeddings, as in $\varphi(x) \otimes \psi(y)$, it denotes the* tensor product of vectors *in the Hilbert space $\mathcal{H}_X \otimes \mathcal{H}_Y$. The meaning of $\otimes$ must therefore be inferred from context, though its use is unambiguous within the proof structure.*

**Theorem 4** (Restatement of Theorem). *Suppose $(\mathcal{X}, d_{\mathcal{X}})$ and $(\mathcal{Y}, d_{\mathcal{Y}})$ are metric spaces of strong negative type. Let $(X, Y, Z)$ be random elements on a common probability space, and define for each $z$ the conditional measures $\mu_z = P_{X|Z=z}$, $\nu_z = P_{Y|Z=z}$ and $P_{(X,Y)|Z=z}$ as above. Then*

$$\mathrm{dCov}^2(X, Y \mid Z) = 0 \quad \Longleftrightarrow \quad P_{(X,Y)|Z=z} \;=\; \mu_z \otimes \nu_z \quad \text{for } P_Z\text{-a.e. } z.$$

*Proof.* Choose isometric embeddings

$$\varphi : \mathcal{X} \hookrightarrow \mathcal{H}_X, \qquad \psi : \mathcal{Y} \hookrightarrow \mathcal{H}_Y$$

into real Hilbert spaces. Thereby automatically follows that each of $\mathcal{X}, \mathcal{Y}$ has strong negative type, and moreover so that each barycenter map $\beta_\varphi, \beta_\psi$ is injective on all finite-moment signed measures (Lyons, 2013, Lem. 3.9). Consider the tensor-product embedding

$$\varphi \otimes \psi : \; \mathcal{X} \times \mathcal{Y} \;\to\; \mathcal{H}_X \otimes \mathcal{H}_Y,$$

and let $\theta_z := P_{(X,Y)|Z=z}$ be the conditional joint law on $\mathcal{X} \times \mathcal{Y}$, with marginals $\mu_z, \nu_z$.

Analog to Lyons (2013, Prop. 3.7), we have for each $z$ the identity

$$\mathrm{dCov}^2\big(X, Y \mid Z = z\big) \;=\; 4 \left\| \beta_{\varphi \otimes \psi}\big(\theta_z - \mu_z \otimes \nu_z\big) \right\|^2_{\mathcal{H}_X \otimes \mathcal{H}_Y}.$$

Taking expectation in $z \sim P_Z$ gives

$$\mathrm{dCov}^2(X, Y \mid Z) = \mathbb{E}_Z \, \mathrm{dCov}^2\big(X, Y \mid Z = z\big) = 4 \, \mathbb{E}_Z \left\| \beta_{\varphi \otimes \psi}(\theta_z - \mu_z \otimes \nu_z) \right\|^2.$$

Since norms are nonnegative and barycenters are well-defined on all finite-moment signed measures, it follows that

$$\mathrm{dCov}^2(X, Y \mid Z) = 0 \quad \Longleftrightarrow \quad \beta_{\varphi \otimes \psi}(\theta_z - \mu_z \otimes \nu_z) = 0 \quad P_Z\text{-a.s.}$$

Injectivity of $\beta_{\varphi \otimes \psi}$ (Lyons, 2013, Lem. 3.9) on signed measures then yields

$$\theta_z - \mu_z \otimes \nu_z = 0 \quad \Longleftrightarrow \quad \theta_z = \mu_z \otimes \nu_z,$$

for $P_Z$-almost every $z$, as required. □

### E.3 DISCO$_m$ PROOF SKETCH

**Proposition 5** (Consistency of DISCO$_m$). *Let $\{(X_i, Y_i, Z_i)\}_{i=1}^n \overset{i.i.d.}{\sim} P_{XYZ}$ with finite first moments, and let $K_h$ be a positive-definite kernel on $\mathcal{Z}$ with bandwidth $h \to 0$, such that the classical kernel regression assumptions hold:*

- *$K$ is bounded, continuous, and integrates to 1,*

- *$h \to 0$ and $nh^{d_Z} \to \infty$ as $n \to \infty$.*

*Then, for each fixed $m$ (or for $m = n$),*

$$DISCO_m(X, Y \mid Z) \overset{p}{\to} \mathrm{dCor}^2(X, Y \mid Z).$$

*Proof Sketch.* **Step 1. Population quantity.** The conditional distance covariance is defined as

$$\mathrm{dCov}^2(X, Y \mid Z = z) = \mathbb{E}[a(X, X')\, b(Y, Y') \mid Z = z],$$

where $(X', Y')$ is an independent copy of $(X, Y)$ given $Z = z$. Thus the global target is

$$\mathrm{dCor}^2(X, Y \mid Z) = \mathbb{E}_Z\big[\mathrm{dCor}^2(X, Y \mid Z = z)\big].$$

**Step 2. Kernel regression approximation.** For a fixed reference point $Z_i$, the kernel weights

$$w_k^{(i)} = \frac{K_h(Z_i, Z_k)}{\sum_\ell K_h(Z_i, Z_\ell)}$$

form a Nadaraya–Watson estimator of the conditional distribution around $Z_i$. Hence, for any integrable function $f$,

$$\sum_k w_k^{(i)} f(X_k, Y_k) \overset{p}{\to} \mathbb{E}[f(X, Y) \mid Z = Z_i].$$

**Step 3. Local centered distances.** The locally centered matrices $A_{k\ell}^{(i)}, B_{k\ell}^{(i)}$ converge to the conditionally centered distances $a(X_k, X_\ell \mid Z = Z_i)$ and $b(Y_k, Y_\ell \mid Z = Z_i)$, respectively.

**Step 4. Local conditional covariance.** The DISCO$_m$ local estimator at $Z_i$,

$$\widehat{\mathrm{dCov}}_{DISCO}(X, Y \mid Z_i) = \frac{1}{\sum_{k,\ell} w_{ik} w_{i\ell}} \sum_{k,\ell} w_{ik} w_{i\ell} A_{k\ell}^{(i)} B_{k\ell}^{(i)},$$

therefore converges to $\mathrm{dCov}^2(X, Y \mid Z = Z_i)$.

**Step 5. Averaging.** Averaging over $m$ reference points yields

$$DISCO_m(X, Y \mid Z) = \frac{1}{m} \sum_{i=1}^m \frac{\widehat{\mathrm{dCov}}_{DISCO}(X, Y \mid Z_i)}{\sqrt{\widehat{\mathrm{dCov}}_{DISCO}(X, X \mid Z_i)\, \widehat{\mathrm{dCov}}_{DISCO}(Y, Y \mid Z_i)}},$$

which converges in probability to the expectation over $Z$, i.e. $\mathrm{dCor}^2(X, Y \mid Z)$.

**Step 6. Ratio convergence.** The denominators are consistent by the same argument, and the ratio converges by Slutsky's theorem.

$\square$

### E.4 sDISCO ESTIMATOR IS CONSISTENT

**Proposition 6** (Consistency of sDISCO). *Let $(X_i, Y_i, Z_i)_{i=1}^n \overset{i.i.d.}{\sim} P_{XYZ}$ with finite first moments, and let $K_h$ be a positive-definite kernel on $\mathcal{Z}$ with bandwidth $h \to 0$ such that the standard assumptions of kernel regression hold (bounded, continuous kernel, $nh^{d_Z} \to \infty$, etc.). Then*

$$sDISCO(X, Y \mid Z) \overset{p}{\to} \mathrm{dCor}^2(X, Y \mid Z),$$

*i.e., the sDISCO estimator is consistent for conditional distance correlation.*

*Proof Sketch.* **1. Target quantity.** The population conditional distance covariance can be written in kernel-weighted form (see e.g. Székely–Rizzo (2014), Sejdinovic et al. (2013)):

$$\mathrm{dCov}^2(X, Y \mid Z) = \mathbb{E}_Z\Big[\mathrm{dCov}^2(X, Y \mid Z = z)\Big].$$

Using kernel regression with weights $w_{ij} = K_h(Z_i, Z_j)$, one approximates conditioning on $Z$ by smoothing over neighbors in $Z$-space.

**2. Difference between DISCO$_m$ and sDISCO.**

- **DISCO$_m$:** fixes a reference $Z_i$, applies local kernel regression around $Z_i$, then averages across $i$.

- **sDISCO:** symmetrizes over all pairs $(i, j)$, mixing reference points. This is equivalent to computing the *U-statistic* version of the same kernel regression estimator.

Formally, sDISCO replaces the empirical conditional centering operator (local per-$i$) by a globally weighted version:

$$A_{ij} = a_{ij} - \bar{a}_{i\cdot} - \bar{a}_{\cdot j} + \bar{a}_{\cdot\cdot},$$

with kernel weights ensuring that only pairs with close $Z_i \approx Z_j$ contribute significantly.

**3. Global centering recovers conditional centering.** Note that in the population limit:

$$\bar{a}_{i\cdot} \approx \mathbb{E}[d_{\mathcal{X}}(X_i, X) \mid Z_i], \quad \bar{a}_{\cdot j} \approx \mathbb{E}[d_{\mathcal{X}}(X, X_j) \mid Z_j],$$

and

$$\bar{a}_{\cdot\cdot} \approx \mathbb{E}[d_{\mathcal{X}}(X, X') \mid Z, Z'],$$

where $(X', Z')$ is an i.i.d. copy. Thus the global centering applied to each pair $(i, j)$ yields exactly the conditional double-centering of distance matrices at the population level. The same holds for $B_{ij}$ in $\mathcal{Y}$.

**4. Law of large numbers with kernel smoothing.** Under the standard kernel regression assumptions:

$$\frac{1}{S} \sum_{i,j} w_{ij} A_{ij} B_{ij} \quad \xrightarrow{p} \quad \mathbb{E}\big[\mathrm{dCov}^2(X, Y \mid Z)\big].$$

This is because the kernel weights $w_{ij}$ concentrate on pairs with $Z_i \approx Z_j$, so the weighted empirical averages converge to conditional expectations. Symmetrization ensures the centering terms cancel correctly, giving an unbiased U-statistic–type estimator, that implicitly sums over all different reference points giving a global measure matching the DISCO variant.

**5. Normalization (correlation).** The same argument applies to the denominators $\mathrm{dCov}^2(X, X \mid Z)$ and $\mathrm{dCov}^2(Y, Y \mid Z)$. By Slutsky's theorem, the ratio converges to the true conditional correlation.

$\square$

**Conclusion.** sDISCO is simply the globally symmetrized, U-statistic version of DISCO$_m$. Since DISCO$_m$ is known to be consistent under kernel regression assumptions, and the centering in sDISCO yields the same conditional expectations in the population limit, sDISCO is also consistent. The advantage is computational: instead of averaging over local references, sDISCO achieves the same limit in one global computation.

# F EXPERIMENTS

## F.1 DATASETS

### F.1.1 CUSTOM DSPRITES

We implemented our own version of the dSprites dataset (Matthey et al., 2017) that allows for full control of the shapes. We control the shape, orientation, x/y-positions, and the scale of objects placed within a 2D grid.

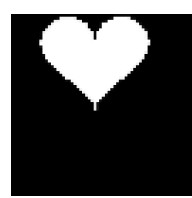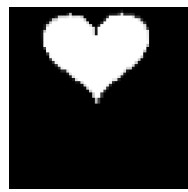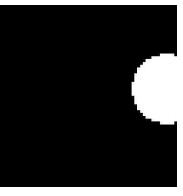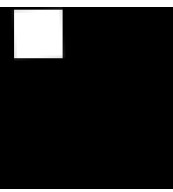

Figure F.3: Custom dSprites examples.

### F.1.2 BLOB DATASET

Inspired by Adeli et al. (2021), we generate synthetic images with two Gaussian blobs. We control the intensities of the blobs, and bias their relationships as described in the main body.

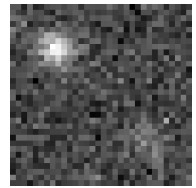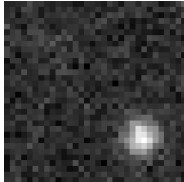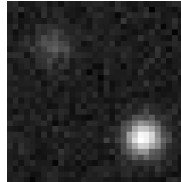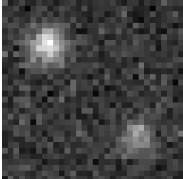

Figure F.4: Blob dataset examples.

### F.1.3 YALEB.

The extended YaleB dataset (Georghiades et al., 2001; Yale, 2001) gives us images of faces in different poses. Additionally, these images have a single light source that is varied across different combinations. We utilize the azimuth and elevation of the light source as a bias as described in the main body of this manuscript.

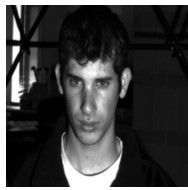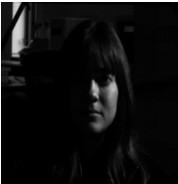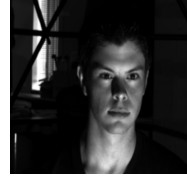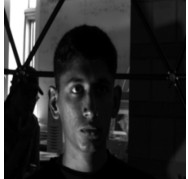

Figure F.5: Extended yaleB examples

### F.1.4 FAIRFACE

The FairFace dataset (Karkkainen & Joo, 2021) provides a range of demographic annotations that can be leveraged to study and mitigate bias. In our experiments, we focus on sex and skin tone as salient attributes. Since FairFace does not provide an explicit skin tone annotation, we inferred it by reinterpreting the dataset's "race" attribute, specifically the "Black" label, which in

practice corresponds most consistently to darker skin tones. Importantly, we do not endorse the conceptualization of "race" as used in FairFace: the category is ontologically unstable, lacks scientific grounding, and reproduces historically problematic constructs.

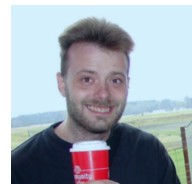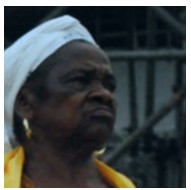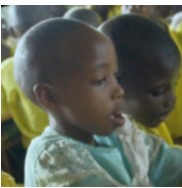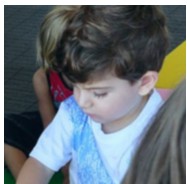

Figure F.6

### F.1.5 WATERBIRDS

We adopt the Waterbirds dataset (Sagawa et al., 2019), constructed by overlaying birds (Welinder et al., 2010; Wah et al., 2011) on land or water backgrounds (Zhou et al., 2017).

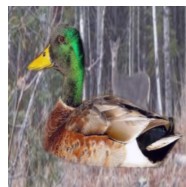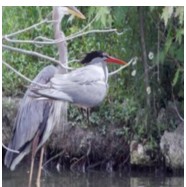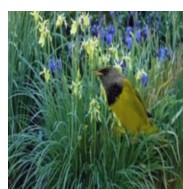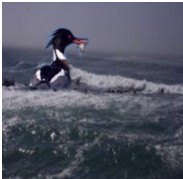

Figure F.7

### F.1.6 MNLI

**Dataset and Experimental Setup.** We use the MNLI dataset (Williams et al., 2018), adopting the exact bias mitigation setup described by Sagawa et al. (2019). The task is to predict the relationship $\in \{entailment, neutral, contradiction\}$ between a sentence pair, where negation terms (e.g., 'nobody', 'no', 'never', 'nothing') act as spurious features highly correlated with the target label.

Unlike our other datasets (Waterbirds, dSprites, Blob, FairFace, YaleB), which offered enough control or balance to construct unbiased validation and test sets, the MNLI base dataset is heavily biased. Since creating a strictly unbiased test set would require discarding excessive data, we follow the protocol of Sagawa et al. (2019): we report Worst Group Accuracy (WGA) and perform model selection based on this metric. Note that WGA is only applicable to datasets where both targets and bias attributes are **categorical**.

**Experimental Results.** For this task, we utilize TinyBERT (Jiao et al., 2020) as the backbone. Tab. 3 demonstrates that the DISCO variants remain highly competitive. GDRO was the only method to outperform DISCO$_m$, doing so by a narrow margin of 0.4% WGA. Conversely, the Adversarial and CIRCE methods proved unstable, performing significantly worse than the naive baseline (TinyBERT without regularization), despite having access to the same hyperparameter search space.

We conclude that our proposed sDISCO and DISCO$_m$ predictors are highly flexible across variable types and data domains (image and text). Our methods consistently lead the performance tables or, at worst, rank second.

### F.2 BIAS MITIGATION METHODS

All of the baseline methods apply different ideas and techniques, but need a backbone architecture. We used the same architecture, depending on the dataset, for all bias mitigation methods. We utilized the classical ResNet (He et al., 2016) in all experiments. To be more detailed, we used the ResNet-18

| Model | WGA | BAcc |
|---|---|---|
| TinyBERT | $0.616 \pm 0.010$ | $0.785 \pm 0.005$ |
| Adversarial | $0.356 \pm 0.058$ | $0.731 \pm 0.010$ |
| eDISCO | $0.730 \pm 0.020$ | $0.778 \pm 0.005$ |
| $DISCO_m$ | $\underline{0.751 \pm 0.007}$ | $0.784 \pm 0.009$ |
| CIRCE | $0.516 \pm 0.069$ | $0.764 \pm 0.009$ |
| C-MMD | $0.627 \pm 0.007$ | $0.785 \pm 0.004$ |
| Fishr | $0.645 \pm 0.017$ | $0.772 \pm 0.003$ |
| GDRO | $\mathbf{0.755 \pm 0.010}$ | $0.785 \pm 0.000$ |
| HSCIC | $0.716 \pm 0.020$ | $0.791 \pm 0.001$ |
| IRM | $0.738 \pm 0.008$ | $0.789 \pm 0.003$ |

Table 3: Results of methods on MNLI. Reported are worst group accuracy (WGA) and balanced accauracy (BAcc).

variant implemented in the torchvision library (maintainers & contributors, 2016), pretrained on ImageNet (IMAGENET1K_V1)(Deng et al., 2009), for dSprites, yaleB, Waterbirds, and FairFace. For the Blob dataset, which deals with smaller images, we used our own implementation of the ResNet architecture that uses group norm (GN) instead of batch norm, as GN was more stable in our experiments. We also changed the depth to two residual blocks, and edited the maxpooling logic to fit the requirements for this small resolution dataset.

For MNLI, we used the original TinyBERT of Jiao et al. (2020) and directly used their officially pre-trained version from HuggingFace at `https://huggingface.co/huawei-noah/TinyBERT_General_4L_312D`.

All details can be found in our repository.

### F.2.1 GDRO

We implemented GDRO exactly following the code and paper by Sagawa et al. (2019). GDRO is one of the standard methods when it comes to bias mitigation as it directly builds on the DRO method that can be seen as a defacto standard in domain generalization (Ben-Tal & Nemirovski, 2002). GDRO is a very efficient method that does not introduce any additional hyperparameters in its standard formulation, compared to the naive ResNet.

### F.2.2 ADVERSARIAL

There are many different suggestions how to design adversarial networks for domain generalization or bias mitigation (Ganin et al., 2016; Wang et al., 2019; Adeli et al., 2021). But most of them differ in some details rather than the general approach. The adversarial network, as all other methods in this paper as well, use the ResNet backbone. Following the ResNet activations, we attach different heads. One head will always be responsible for the task prediction, thus predicting the target $Y$. Then, for every bias variable present, a new head will be introduced. The optimization procedure is as follows:

- predict the task. Update ResNet and the task head parameters.
- for each bias, predict it. Update only the bias prediction heads (protected attribute step).
- for each bias, predict it. Use the negative gradient of the loss and update only the ResNet (unlearn step).

There are many different ways to parametrize the optimization procedure for this adversarial setting. Some suggest using different learning rates for each of the network parts, including each of the bias heads. We, instead, use a single learning rate for the entire network with all of its parts. To adjust the weighting of the different network parts, we use $\lambda_{protected}$ as the parameter that controls the learning weight for the task head. All bias heads use the same weighting parameter $\lambda_{unlearn}$. Instead of using different learning rates, this loss weighting reduces to an equivalent formulation, while being more comparable to the other, regularzation-based methods. See Tab. 4 for the search space.

### F.2.3   C-MMD

We implemented c-MMD, as formulated similarly by Makar et al. (2022); Kaur et al. (2022); Veitch et al. (2021). It tries to penalize predictions or latent representations based on conditional MMD. Besides the penalty strength $\lambda$, it introdcues a single bandwidth (bw) hyperparameter that controls the (Gaussian) kernel's bandwidth on the network prediction outputs. See Tab. 4 for the search space.

### F.2.4   HSCIC

This method is an direct extension of the classical HSIC measure (Gretton et al., 2005; 2007) to the conditional setting. Quinzan et al. (2022) first used it in their work, but their implementation is extremely inefficient. We directly adapted the efficient implementation given by Pogodin et al. (2022).

This method is the most expensive one, when it comes to hyperparameters. It introduces 5 hyperparameters. We did a grid search over 4 of them, as this space was already immense, compared to the others, and held one of them fixed, based on initial results. The 4 hyperparameters we searched over is the regularization strength, $\lambda$, and three bandwidth parameters, $\sigma$, for each of the involved variables in the conditional independence criterion. It further has another parameter, the ridge regression regularization parameter, $\lambda_{ridge}$, we which held fixed at 0.01, following similar values as in Pogodin et al. (2022); Quinzan et al. (2022), and also ensuring in preliminary runs that this value is advantageous for the method in our settings. See Tab. 4 for the search space.

### F.2.5   CIRCE

CIRCE (Pogodin et al., 2022) is a direct extension of the HSCIC method. It was published as a direct successor or competitor. CIRCE, in theory, has a similar set of hyperparameters. But it reduces the search space by doing a small, pre-training hyperparameter optimization that does not depend on the neural network, in advance. See Tab. 4 for the search space.

### F.2.6   DISCO

Our methods, DISCO$_m$ and sDISCO, introduce the regularization strength $\lambda$ and the bandwidth (bw) parameter for the Gaussian kernel. See Tab. 4 for the search space.

### F.2.7   HYPERPARAMETERS

Table 4: Hyperparameter grids searched for each method, along with the total number of runs evaluated.

| Method | Hyperparameter Grid | Runs |
|---|---|---|
| Adversarial Network | $\lambda_{\text{protected}} \in \{10.0, 5.0, 2.0, 1.0, 0.5, 0.1\}$
$\lambda_{\text{unlearn}} \in \{10.0, 5.0, 2.0, 1.0, 0.5, 0.1\}$ | $6 \times 6 = 36$ |
| DISCO (both) | bw $\in \{1.0, 0.9, 0.5, 0.1, 0.01, 0.001\}$
$\lambda \in \{10.0, 5.0, 2.0, 1.0, 0.5, 0.1\}$ | $6 \times 6 = 36$ |
| CIRCE | $\lambda \in \{1000.0, 100.0, 10.0, 1.0, 0.1\}$
$\sigma_y^2 \in \{0.9, 0.5, 0.1, 0.01, 0.001\}$
$\sigma_z^2 \in \{0.9, 0.5, 0.1, 0.01, 0.001\}$ | $5 \times 5 \times 5 = 125$ |
| c-MMD | bw $\in \{1.0, 0.9, 0.5, 0.1, 0.01, 0.001, 0.0001, 0.00001\}$
$\lambda \in \{1000.0, 100.0, 10.0, 1.0, 0.5, 0.1\}$ | $8 \times 6 = 48$ |
| HSCIC | $\lambda \in \{1000.0, 100.0, 10.0\}$
$\sigma_y^2 \in \{0.5, 0.1, 0.01, 0.001\}$
$\sigma_z^2 \in \{0.1, 0.01, 0.001\}$
$\sigma_y^2 \in \{0.1, 0.01, 0.001\}$ | $3 \times 4 \times 3 \times 3 = 108$ |
| IRM | $\lambda \in \{1.0, 2.0, 10.0, 100.0, 500.0, 1000.0, 5000.0, 10000.0, 50000.0, 100000.0\}$
warm-up epochs $\in \{0, 10, 30, 40\}$ | $10 \times 4 = 40$ |
| Fishr | $\lambda \in \{1.0, 2.0, 10.0, 100.0, 500.0, 1000.0, 5000.0, 10000.0, 50000.0, 100000.0\}$
warm-up epochs $\in \{0, 10, 30, 40\}$ | $10 \times 4 = 40$ |

### F.3 Path Analysis Counterfactuals

As a recap, for this experiment, scale is our target attribute, and we predict it in a binary manner (large vs. small scale). The $X$ and $Y$ positions of the object itself work as biases we want to mitigate.

In the manuscript, we quantified model sensitivity to counterfactual changes of a bias variable $X$ via the following measure:

$$S_X(\theta) := \mathbb{E}_{u \sim U} \, \mathbb{E}_{x \sim \mathrm{Unif}(\mathrm{supp}(X))} \left[ \left| P(\hat{Y}(u); \theta_m) - P(\hat{Y}_x(u); \theta_m) \right| \right],$$

and analogously defined $S_Y(\theta)$ for the variable $Y$.

In addition, we further introduced counterfactual accuracy,

$$Acc_{ctf} := \mathbb{E}_{u \sim U} \, \mathbb{E}_{s \sim \mathrm{Unif}(\mathrm{supp}(Scale))} \left[ \mathbf{1}\big\{ Y(u) = \hat{Y}_s(u) \big\} \right],$$

which evaluates whether predictions remain consistent with ground-truth outcomes under counterfactual changes of the causally relevant target variable Scale.

To visually understand what these formulas measure, we fix a single unit $u \in U$. The following Fig. F.8 shows what some of the different counterfactuals look like for this given unit $u$.

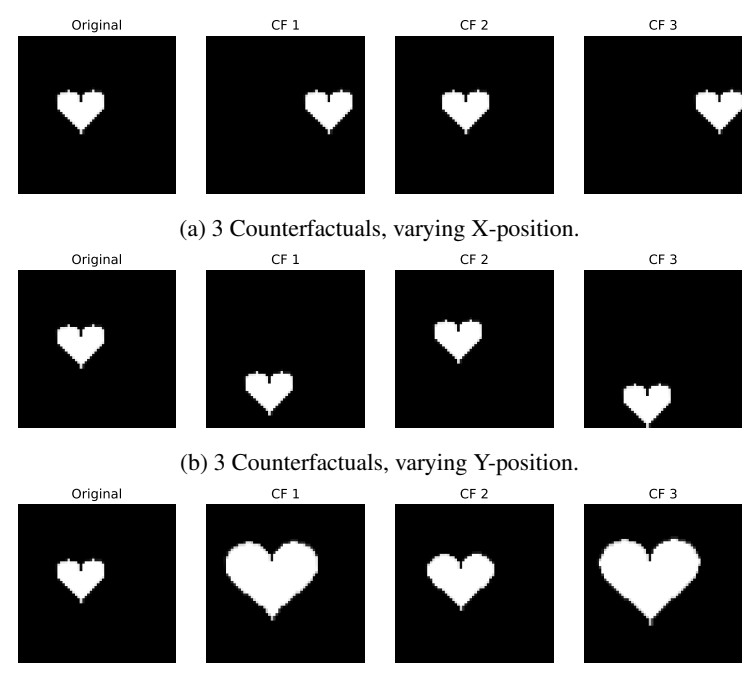

(a) 3 Counterfactuals, varying X-position.

(b) 3 Counterfactuals, varying Y-position.

(c) 3 Counterfactuals, varying Scale.

Figure F.8: Counterfactual images for a given unit $u$, labeled here as "Original".

### F.4 Counterfactual Analysis on dSprites

To further illustrate the benefits of counterfactual analysis, we consider the same dSprites setup as in the main manuscript (Sec. 4). Specifically, we evaluate one of the already hyperparameter-optimized, best-performing variants for each method.

In the main experiments, we followed standard practice in the domain generalization literature and reported results on bias-balanced validation and test sets. While this setting is appropriate for assessing overall causal stability in a single balanced setting, it does not necessarily reveal *why* a model fails. In particular, causal stability is not the only goal. While a model might be stable to counterfactual changes of the bias attribute, it might do so by also reducing overall predictive performance.

Counterfactual analysis, grounded in the causal graph formalized through SAM, provides these exact complementary insights. To recall the setup from the Section 4: the target variable $Y$ is the vertical position (y-coordinate) of the object in the 2D plane, while the bias attribute is the horizontal position (x-coordinate), denoted as $X$.

Analogous to pathway analysis, we define

$$S_X(\theta) := \mathbb{E}_{u \sim U} \, \mathbb{E}_{x \sim \text{Unif}(\text{supp}(X))} \left[ \left| \hat{Y}(u) - \hat{Y}_x(u) \right| \right],$$

which measures the path-specific sensitivity of the model output with respect to the attribute $X$.

Similarly, in analogy to counterfactual accuracy, we introduce the *counterfactual $R^2$ score*:

$$R^2_{\text{ctf}} := 1 - \frac{\mathbb{E}_{u \sim U} \, \mathbb{E}_{y \sim \text{Unif}(\text{supp}(Y))} \left[ \left( y - \hat{Y}_y(u) \right)^2 \right]}{\mathbb{E}_y \left[ \left( y - \bar{Y}(u) \right)^2 \right]},$$

where $\bar{Y}$ denotes the mean of $Y$ over all counterfactually generated samples. This parallels the standard $R^2$ score, but is evaluated under counterfactual interventions on the $Y$ attribute.

Table 5: Counterfactual sensitivity $S_X$ and counterfactual $R^2$ scores across methods on dSprites. Lower $S_X$ indicates reduced spurious dependence on the bias attribute $X$, while higher $R^2_{\text{ctf}}$ indicates better counterfactual predictive performance.

| **Method** | $S_X$ | $R^2_{\text{ctf}}$ |
|---|---|---|
| $\text{DISCO}_m$ | 0.058 | 0.7181 |
| sDISCO | 0.067 | 0.6474 |
| HSCIC | 0.064 | 0.6021 |
| CIRCE | 0.064 | 0.5939 |
| Adversarial | 0.084 | 0.4812 |

Since $Y$ has a standard deviation of 0.2 in the test set, we observe that all models manage to reduce dependence on the bias variable $X$, though to varying degrees. Recall from Tab. 2 in the main text that sDISCO ranked above HSCIC and CIRCE in terms of balanced accuracy. While this metric reflects a combination of debiasing and predictive performance, it does not disentangle which factor contributed more.

Counterfactual analysis provides this finer-grained perspective. Tab. 5 shows that sDISCO, HSCIC, and CIRCE all achieve comparable mitigation of bias effects, as indicated by similar $S_X$ values. However, their counterfactual $R^2$ scores reveal a different story: both HSCIC and CIRCE reduce dependence on $X$ at the cost of predictive accuracy on the task-relevant variable $Y$.

In summary, while balanced accuracy on a bias-balanced test set is useful for reporting overall performance, counterfactual path analysis enabled by SAM allows us to better understand *why* certain models fail. In this experiment, we see that HSCIC and CIRCE, although nearly matching sDISCO in debiasing effectiveness, sacrifice too much information from the true target variable $Y$.

