# OpenReview forum: "DISCO: Mitigating Bias in Deep Learning with Conditional Distance Correlation"
_ICLR.cc/2026/Conference — Submitted to ICLR 2026_

### Official Review · Reviewer_FXqj · 2025-10-27

**Soundness:** 3
**Presentation:** 2
**Contribution:** 3
**Rating:** 6
**Confidence:** 3

**Summary:**

The authors consider a 'Standard Anti-Causal Model' which is defined as a causal graph where predictions $\hat{Y}$ are derived from observed features X, which in turn can be influenced by confounding, collider variables, and the true label Y. (The label Y causes X, thus a predictor for $\hat{Y}$ makes predictions in the anti-causal direction). For predicting the true label Y, the authors suggest to only consider direct causal effects, without the influence of indirect or biasing factors. This is expressed as a causal stability condition. Eventually, a training goal is formulated as a standard loss minimization problem under the constraint that a to-be-trained model prediction becomes independent of the effects mediated through confounding or collider paths given the true label.

The authors propose a conditional distance correlation measure to approximate the independence of two variables from each other, and approximate the via a kernel regression from data. In the following two separate "DISCO" methods are defined to approximate the degree of conditional independence between the true and predicted target, given the biasing features.

Experiments are conducted on 5 (semi-)synthetic datasets, featuring different types of confounding and collider bias and generally seem to show the well working of the method and against several existing baselines.

**Strengths:**

The paper presents a novel approach for regularizing arbitrary ML model trainings towards unbiased predictions under the influence of biasing factors. While prior works tackled similar problem settings, the paper seems to generalize beyond to a broader class of causal structures that all can be handled with the proposed method. The integration of a causal stability criterion via an addition independence constraint might integrate well within existing setups and makes the method agnostic to the underlying ml model.

The paper is generally well motivated and structured. The presented derivations and proofs are mostly self-contained. The provided figures strongly support understanding of the required graphical criteria and the appendix contains detailed derivation of the individual propositions and theorems.

While the authors consider a particular type of semi-synthetic datasets, their method seems to excel at those in comparison to existing approaches.

**Weaknesses:**

**1) Related Work and Problem Setup.** While the authors cite and compare to several works throughout their paper and in experiments, no explicit related work section is provided. It is therefore difficult to pinpoint qualitative differences to existing approaches.

Given that the authors consider high-dimensional image data during their experiments, no reference to causal representation learning (CRL) literature is made. The presented task, however, seems to slightly differ to common image classification and CRL setups in that the biasing factors B seem to be assumed to already have been annotated (or are otherwise known) for all individual samples. The authors should be more explicit about this requirement, as such additional information is generally not provided for standard image datasets, but arguably simplifies the problem.


**2) Counterfactual Loss and Sufficient Variance. **The authors motivate the concept of causal stability via the optimization of counterfactual effects. On one hand, counterfactual data can commonly not be observed for real-world data. The authors could improve the paper by stating more explicitly which assumptions on the availability of data is made in the experiments to evaluate CF effects, or how to otherwise interpret the meaning of a counterfactual analysis in this work.

On the other hand, the optimization problem of eq. 8 and its realization in eq. 24, purely optimize the model by minimizing a loss via the model output. Common confounding, (e.g., camel<->desert, cow<->meadow), can usually not be broken for associational learners using a loss regularization, as this is caused by a lack of variation in the input. Works in the field of CRL therefore commonly assume sufficient variation of the data, assuming support over the whole distribution (e.g. occasional camel<->meadow samples...). From reading the paper, it is unclear to me whether the authors initially assume sufficient variability or whether the DISCO regularization terms exposes/requires the model to such additional data.


**3) Clarity.** Sec. 3.1 is rather dense in theory, but lacks in explanations on the particular choices and derivation made to arrive at the specific equations given in definitions 3 and 4. In particular, definition 3 is given without any further comment. Here, the the semantics of the additional "a" and "D" terms are unclear to me. The paper might be improved by providing further explanations on how to interpret the given terms in eq. 10.

Similarly, the authors use their distance correlation measures to construct their DISCO losses. While the overall derivation seems to be technically sound, the authors might improve this part by briefly discussing the qualitative differences between $DISCO_m$ and sDISCO estimation.


**Minor:**

* The authors could be more explicit on the role of the "standard assumptions of kernel regression", as mentioned in proposition 2, towards the well working or non-working of their method. I assume this relates to common smoothness conditions of the approximated function?
* Given the strong implications of the assumed causal graph towards identifiability of causal effects, the paper might be improved by moving a variant of figure B.1 directly into the main text.
* Some citations in the proof sketch of proposition 6 are not linked properly. Székely–Rizzo (2014) is therefore missing in the references.

**Questions:**

My questions primarily regard the above weaknesses:

1) Could be authors elaborate on the relation of their problem setup to that of common CRL tasks? How is the method contrasted to existing causal representation learning methods and how realistic is the assumption of having access to annotations on the biasing features, e.g., in common image datasets?
2) Which assumptions on the availability and sufficient variability of the data are made in the paper? Does the DISCO regularization impose additional requirements on the availability of the data?
3) Could the authors elaborate on how to interpret the role of the individual a and D terms in eq. 10? Similarly, the authors construct a new conditional distance measure in eq. 10. Why is this new measure required? E.g., how does it compare to existing ones like HSIC that equally construct kernels?
4) Could the authors elaborate on the assumptions of kernel regression required for their method? Often times kernel methods struggle to approximate non-smooth functions. Is this the case here and would it impact performance on discrete feature spaces?

---

> ### Author Response · Authors · 2025-11-14
>
> We thank the reviewer for this **excellent and insightful question**. It highlights the need to clarify the precise positioning of our work, which we will do in a dedicated related work section in the camera-ready version. Given the page limitations, we prioritized our theoretical (SAM) and practical (DISCO) contributions, but we agree this contextualization is crucial.
>
> # 1 CRL, related work, and realism of known attributes
>
> ---
>
> ## Relation to Common CRL Tasks
>
> We absolutely categorize our work as a **Causal Representation Learning (CRL)** method. CRL is a broad field, but a central goal, as articulated by Schölkopf et al. (2021), is to learn representations that are **robust** and generalize to new environments, often by leveraging causal principles. Our work fits squarely within this goal.
>
> The reviewer asks about "common" CRL tasks. This field is diverse, with methods focusing on different, non-exclusive goals:
> * **(a) Generative & Disentangled CRL:** Learning the full causal *generative* process or disentangling independent causal mechanisms (e.g., [6], [7]).
> * **(b) Latent Confounder CRL:** Inferring causal effects or learning robust representations when key confounders are *unobserved* (e.g., [2], [3], [4]).
> * **(c) Causal Discovery CRL:** Inferring the causal graph structure itself from high-dimensional, observational data (e.g., [5]).
> * **...**
> ---
> ## Contrast with Existing CRL Methods
>
> Our method's primary contribution is different. **We propose a discriminative CRL method** focused on learning a **causally stable predictor** in the presence of *known* bias attributes.
>
> * **What we don't do:** We do not attempt to learn the full generative process (a) or the causal graph (c). We also assume the bias $B$ is *observed*, placing us in a different setting than (b).
> * **What we do:** Our core contribution is to show that for the specific, important problem of bias mitigation, we can *avoid* the complexity of full generative modeling.
>     1.  We first propose a causal framework, the **Standard Anti-Causal Model (SAM)**, to formally model the data-generating process and its biases ($B$).
>     2.  From SAM, we derive a formal, observational criterion for robustness: **Causal Stability** (Definition 1), which is achieved when $\hat{Y} \perp B \mid Y$. We prove this criterion ensures that **spurious ($ctf-SE$)** and **indirect ($ctf-IE$)** effects vanish, isolating the stable direct effect (Theorem 1, Corollary 1).
>     3.  Our method, **DISCO** ($DISCO_{m}$ and sDISCO), is a novel and efficient regularization technique (Equation 24) that allows a deep model to be optimized for precisely this causal criterion.
>
> In short, our method uses causal principles to learn a robust representation (the latent space of the neural network) that is **provably stable** against spurious correlations from $B$, by enforcing a targeted **conditional independence** objective.
>
> ---
>
> ## Realism of Assuming Bias Annotations
>
> The reviewer raises a critical, practical point on the realism of having access to annotations for biasing features ($B$). We agree this is a key consideration.
>
> * **Fairness and Auditing:** This assumption is most realistic in **fairness-critical applications** (e.g., medicine, hiring, finance). In these domains, attributes like age, sex, or clinic location are often collected for auditing and to satisfy legal requirements, even if they are not intended to be used for prediction.
> * **Scientific and Industrial Data:** In many scientific or industrial settings, metadata about the data collection process (e.g., sensor type, lighting conditions, time of day) is available. These variables are common sources of bias and fit our assumption perfectly.
> * **Standard Benchmarks:** The assumption is **standard for the entire field of bias mitigation**. Benchmarks like **Waterbirds** (background bias), **FairFace** (skin tone bias), and **YaleB** (lighting bias) are all built on this premise, as they are explicitly designed to model real-world scenarios where a *known* spurious correlation exists.
>
> We want to highlight that bias and shortcut mitigation are highly established fields in deep learning. This also explains our main positioning of our work within this context, and the choice of our baseline methods. We would love to add CRL contextualization into the intro if the reviewer thinks this improves our work.
>
> [1] Towards Causal Representation Learning, Schölkopf et al. 2021
>
> [2] Vision-and-Language Navigation via Causal Learning, Wang et al. 2024
>
> [3] Causal Effect Inference with
> Deep Latent-Variable Models, Louizos et al.
>
> [4] Deep Learning Methods for Proximal Inference via
> Maximum Moment Restriction, Kompa & Bellamy et al.
>
> [5] Deep learning of causal structures in high dimensions under data limitations, Lagemann et al.
>
> [6] Diffusion Causal Models for Counterfactual Estimation, Sanchez et al.
>
> [7] Learning Causally Disentangled Representations via
> the Principle of Independent Causal Mechanisms, Komanduri et al.

---

> ### Author Response · Authors · 2025-11-14
>
> We sincerely thank the reviewer for this insightful and valuable question. It highlights a crucial, often implicit, assumption in the field of Causal Representation Learning and bias mitigation, and we appreciate the opportunity to clarify our work's position on these two distinct points: 1) counterfactual evaluation and 2) sufficient data variability.
>
> # 2. Sufficient Variability, additional requirements of DISCO, Counterfactual Evaluation Metrics
>
> **a) On Counterfactual (CF) Data for Training vs. Evaluation**
>
> We want to state emphatically that our training method, DISCO, **does not require access to counterfactual data to be trained.**
>
> The reviewer is correct that counterfactual data is generally unobservable. Our entire theoretical motivation is to find an *observational criterion* that ensures causal stability. We show in Theorem 1 that the observational conditional independence $\hat{Y} \perp B | Y$ is sufficient to achieve causal stability (i.e., vanishing counterfactual indirect and spurious effects).
>
> Our proposed DISCO loss (Eq. 24) is an estimator for this *observational* criterion. The optimization problem is:
>
> $$
> \min_{\theta}\sum L(Y,\hat{Y})+\lambda DISCO_{m}(\hat{Y},B|Y)
> $$
>
> It operates purely on the observed data tuples $(X, Y, B)$ available in each training batch.
>
> The confusion, as the reviewer notes, may arise from our **evaluation** in Section 4.4 (Path Analysis). The counterfactual metrics reported there ($S_X$, $S_Y$, $Acc_{ctf}$) were possible **only because we used synthetic datasets (dSprites, Blob)**. In these specific experiments, we have full control over the Structural Causal Model (SCM) and the data-generating process. This "oracle" access allows us to intervene ($do(\cdot)$) and generate the ground-truth counterfactuals needed for this specific *analysis*.
>
> The purpose of this analysis was not to suggest such evaluation is possible on real-world data, but to *validate* that our observationally-trained model does, in fact, successfully optimize the *causal* objective (i.e., stability to counterfactual interventions). We will revise Section 4.4 to make this distinction between our observational *training method* and this specific, synthetic-only *evaluation* more explicit.
>
> **b) On Sufficient Data Variability (Positivity)**
>
> The reviewer raises a fantastic point regarding sufficient variability. Our method, in line with all observational Causal Representation Learning and shortcut mitigation methods, implicitly relies on an assumption of **sufficient variability** (or *overlap*). This is directly analogous to the **positivity assumption** in causal inference.
>
> Specifically, we assume that for any given value of the target $Y=y$, the data distribution has non-zero probability (i.e., support) over all combinations of the bias attributes $B$. Formally, for all realizations $y$, $b$, we require $P(B=b | Y=y) > 0$. Using the reviewer's excellent example, to break the spurious association `cow <-> meadow`, the dataset *must* contain some counter-examples, however rare, such as `cow <-> desert` or `camel <-> meadow`. (Please see the waterbirds dataset. There we have `waterbird <-> water`, `landbird <-> land`, but obviously with sufficient variability / positivity assumption).
>
> To be perfectly clear, the DISCO regularization (Eq. 24) **does not** *create* or *require* additional data beyond the assumption of sufficient variability. Rather, it is an objective that *leverages* the existing (and assumed) variability in the observed data. It penalizes the model for statistical dependencies between $\hat{Y}$ and $B$ within each stratum of $Y$. This optimization is only feasible if the data provides sufficient support to estimate these conditional dependencies. If this variability were absent (e.g., $B$ is a deterministic function of $Y$), the confounding is non-identifiable from observational data alone, and no regularization-based method could succeed.
>
> We want to emphasize again that none of the bias mitigation methods, nor the CRL methods work without the positivity / variability assumption. This is no specific limitation to our work.
>
> We appreciate the reviewer for highlighting this. As it is a standard assumption for this entire class of problems, it is often left unstated in the field of bias mitigation. We will add a formal discussion of this sufficient variability/positivity assumption to our paper (e.g., in Section 2.2 on assumptions) to make this requirement explicit.

---

> ### Author Response · Authors · 2025-11-14
>
> Thank you for this opportunity to elaborate on these excellent and insightful questions. Both points touch on the theoretical foundations of our work, and we're happy to provide more intuition.
>
> We will, of course, add these clarifications to the **camera-ready** version, space permitting, as we agree they are valuable for the reader.
>
> ---
>
> # 3. Interpreting Centering and Comparing to HSIC
>
> This is a multi-faceted question. We'll break it down into the interpretation of the terms and the comparison to the Hilbert-Schmidt Independence Criterion ($\text{HSIC}$).
>
> ### 1. Intuition for the $\text{a}$ and $\text{D}$ terms in Equation 10
>
> The terms in Equation 10 are the population-level components for the **double-centering** of a distance matrix. This is a crucial step in distance correlation ($\text{dCor}$) and is directly analogous to **mean-centering** variables before computing standard covariance.
>
> * **Standard Covariance (Analogy):** To compute $\text{Cov}(X, Y)$, we first center the variables: $\mathbb{E}[(X - \mathbb{E}[X])(Y - \mathbb{E}[Y])]$.
> * **Distance Correlation (Our Case):** In our non-parametric, distance-based setting, the "centering" process involves expectations over distances.
>
> Within a conditional slice $Z=z$:
> * $a_{\mu_{z}}(x) = \int d_{\mathcal{X}}(x,x^{\prime})d\mu_{z}(x^{\prime})$: This is the **average distance from a single point $x$ to all other points** drawn from the conditional marginal distribution $\mu_{z} = P_{X|Z=z}$. It represents the "mean" distance relative to $x$.
> * $D(\mu_{z}) = \int\int d_{\mathcal{X}}(x,x^{\prime})d\mu_{z}(x)d\mu_{z}(x^{\prime})$: This is the **overall average distance between any two points** drawn from $\mu_{z}$. It represents the "grand mean" of all distances.
>
> The final term, $d_{\mu_{z}}(x,x^{\prime}) = d_{\mathcal{X}}(x,x^{\prime}) - a_{\mu_{z}}(x) - a_{\mu_{z}}(x^{\prime}) + D(\mu_{z})$, is the **distance-centered kernel**. This "double-centering" ensures that the resulting measure, distance covariance ($\text{dCov}$), is zero if and only if the variables are independent.
>
> **Matrix-Based Intuition (Sample Estimation):**
> In the sample domain, this procedure corresponds precisely to double-centering the pairwise distance matrix $A$, where $A_{ij} = d(x_i, x_j)$.
>
> The correctly centered matrix expression should be represented as:
> $$A_{\mathrm{C}, ij} = A_{ij} - A_{i}^{\mathrm{mean}} - A_{j}^{\mathrm{mean}} + A_{\mathrm{grand}}^{\mathrm{mean}}$$
> where $A_{i}^{\mathrm{mean}}$ is the row mean ($a_{\mu_{z}}(x_i)$ analog) and $A_{\mathrm{grand}}^{\mathrm{mean}}$ is the grand mean ($D(\mu_{z})$ analog). Sorry to change the notation here slightly, the openreview markdown+latex environment did not like the \bar{} statements we used combined with the \cdot.
>
> ### 2. Why is this Measure Required? (vs. HSIC)
>
> We are building upon the established conditional distance correlation measure, with our main contribution being the **scalable, differentiable estimators ($\text{DISCO}_m$ and $\text{sDISCO}$)** that make it viable for optimization in deep learning.
>
> The key insight is found in the foundational work of Sejdinovic et al. (2013): **Distance Covariance ($\text{dCov}$) is mathematically equivalent to a specific form of $\text{HSIC}$**.
> $$\mathcal{V}^2(X,Y) = 4\gamma_k^2(P_{XY}, P_X P_Y)$$
> where $\mathcal{V}^2(X,Y)$ is the squared distance covariance and $\gamma_k^2$ is the squared Maximum Mean Discrepancy ($\text{MMD}$) on the product space (which is $\text{HSIC}$).
>
> The crucial difference is the **implicit kernel** utilized:
>
> * **"Classic" HSIC:** Typically employs **translation-invariant kernels** like the Gaussian (RBF) kernel.
> * **Distance Correlation ($\text{dCor}$):** Implicitly uses a specific **distance-induced kernel**. For $\rho(x, x') = ||x - x'||$, the equivalent kernel is of the form $k(x, x') \propto ||x|| + ||x'|| - ||x - x'||$.
>
> This distance-induced kernel arises because the weighting function in Fourier space for $\text{dCor}$ is non-integrable (violating Bochner's theorem for standard translation-invariant kernels).
>
> **Conclusion:** We are not proposing a fundamentally "new" measure of dependence versus $\text{HSIC}$, but rather leveraging $\text{dCor}$'s inherent use of a **different class of kernels**. Our empirical results (Table 2) show that $\text{DISCO}$ consistently outperforms the $\text{HSCIC}$ baseline, suggesting this specific (implicit) kernel choice is **more effective for the high-dimensional spurious correlation problems** in our bias mitigation tasks.
>
> Furthermore, the distance based measures require less hyperparameters and showed to be much more stable and reliable in our experiments for deep learning optimization.
>
> And, lastly, the concurring methods of HSCIC, or CIRCE need to perform additional complex kernel regression steps. We, instead, proposed to measure the global conditional dependence via NW estimation, and even an efficient one-shot version in sDISCO.

---

> ### Author Response · Authors · 2025-11-14
>
> # 4. Kernel Assumptions and Discrete Spaces
>
> This addresses the theoretical assumptions required for consistency and the practical implications for data types.
>
> ### 1. Assumptions of Kernel Regression
>
> Our method relies on the "standard assumptions of kernel regression" for the consistency of the Nadaraya-Watson kernel estimator, which is used to approximate conditional expectations. These ensure reliable local averaging:
>
> * **Kernel Properties:** The kernel ($K_h$) must be standard, bounded, and continuous.
> * **Bandwidth Control:** The bandwidth $h \rightarrow 0$ as sample size $n \rightarrow \infty$.
> * **Sufficient Data:** The condition $n h^{d_z} \rightarrow \infty$ (where $d_z$ is the dimension of the conditioning variable $Z$) must hold to mitigate the curse of dimensionality.
>
> **Non-smoothness** may affect the convergence *rate* of the estimator, but it does not invalidate the *consistency* of the overall $\text{DISCO}$ measure itself, as our goal is only the consistent estimation of the conditional dependence, not perfect local regression.
>
> ### 2. Impact on Discrete Feature Spaces
>
> We thank the reviewer for this insightful question regarding the behavior of our proposed method, DISCO, in the context of non-smooth functions and discrete feature spaces. This is a critical point, and we are pleased to clarify how DISCO is particularly well-suited for this scenario.
>
> 1.  **dCor Detects All Types of Dependence:** The core strength of distance correlation (and our conditional extension, DISCO) is that it is a measure of dependence based on characteristic functions. As stated in Theorem 3 of our paper, $dCov^{2}(X, Y | Z) = 0$ if and only if $X$ and $Y$ are conditionally independent given $Z$. This property holds true regardless of the underlying functional relationship between the variables, whether it be linear, non-linear, smooth, or non-smooth/discrete. Our goal is not to *approximate* a specific function, but to *detect the presence* of any conditional dependence, for which dCor is expressly designed.
>
> 2.  **Role of the Kernel in DISCO:** In our DISCO estimators (Eq. 17-23), the kernel $K_h$ is applied *only* to the conditioning variable $Z$ (which is the target variable $Y$ in our final objective, Eq. 24). This kernel is used to create weights $w_{ij}$ for a standard non-parametric smoothing (i.e., kernel regression) to estimate the conditional expectation present in the definition of $dCov^2(X,Y|Z)$ (Eq. 11). This is a well-established technique for conditioning. This kernel does *not* define the metric for measuring the dependence between our variables of interest ($\hat{Y}$ and $B$). That dependence is captured by the Euclidean distances $a_{ij}$ and $b_{ij}$ (Eq. 16).
>
> 3.  **Application to Discrete (One-Hot) Spaces:**
>     * For the discrete, one-hot encoded variables used in our experiments (e.g., on FairFace and Waterbirds), the notion of a "smooth function" in the continuous sense does not directly apply.
>     * When DISCO's kernel smoothing is applied to a discrete conditioning variable $Y$, the Gaussian kernel (with an appropriate bandwidth) effectively functions as a "soft" delta kernel. It assigns high weight to samples where the $Y$ categories match ($Y_i = Y_j$) and progressively lower weights as they differ, which is a robust way to approximate the conditioning.
>     * Crucially, the distance correlation itself is perfectly capable of operating on the discrete spaces for $\hat{Y}$ and $B$, as it simply relies on the distances between the output predictions and the (one-hot) bias vectors.
>
> In summary, the reviewer is correct that methods relying on smooth kernels to *define* their dependence metric can struggle with non-smooth functions. However, our method, DISCO, is not one of them. It uses distance correlation, which is theoretically guaranteed to detect non-smooth dependencies, and employs kernels only for the standard and robust task of non-parametric conditioning, whereas HSCIC and CIRCE employ kernels and kernel regression throughout all steps, leading to the high dependence on hyperparameters and failing to provide good solution in some cases. The theoretical suitability is finally empirically validated by our strong performance on datasets with discrete variables, as shown in Table 2. Indeed, our method handles mixed variables, as in yaleB, significantly better than the other methods.

---

> > ### Comment · Reviewer_FXqj · 2025-11-22
> >
> > I thank the authors for their detailed response that clarifies many of the asked questions. The fact that the method relies an a sufficient variability/overlap/positivity in B assumption is reasonable and agree that it is, indeed, generally difficult to proof identifiability without it. Mentioning this assumption in the paper will resolve my point. Although being slightly restrictive, knowledge/annotation of biasing factors, and the use of CF data for testing only, can make sense in the considered setting.
> >
> > I appreciate the detailed explanations w.r.t. Sec. 3.1. The given explanations are quite insightful and help a lot in relating the approach to existing methods. Similarly, remarks on the assumptions of kernel regression and the particularly fit of the method for non-smooth functions are an interesting insight that might be quite relevant for many ML tasks.
> >
> > As also remarked by the other reviewers, the authors operate and test under a setting that imposes quite strong requirements on the available data. While this might be due to the specific overlap of the fields of causality and bias mitigation, experiments use rather simple, or need to modify, datasets (e.g., modifying waterbirds) which lowers the impact of experiments. I'm torn between the fact that causal identification is usually hard to achieve under very general conditions on one hand, and the experiments being quite tailored to the proposed method on the other. Given that I'm already in favor of accepting the paper I will remain with my current score, but might be inclined to raising it further, depending on the outcome of the other discussions.

---

### Official Review · Reviewer_Ys2v · 2025-11-01

**Soundness:** 3
**Presentation:** 3
**Contribution:** 3
**Rating:** 6
**Confidence:** 3

**Summary:**

The paper proposes a causal framework for mitigating bias in deep learning by ensuring that model predictions rely only on causal, stable information rather than spurious shortcuts. Using the Standard Anti-Causal Model (SAM), the authors show that achieving causal stability amounts to enforcing $\hat{Y}\perp B | Y$, i.e., models' predictions  $\hat{Y}$ should be independent of the spurious bias variable $B$ when conditioned on the true label $Y$, which is treated as the cause in this paper.

To implement the idea, they introduce two estimators of conditional distance correlation. $DISCO_m$ and $sDISCO$, which quantitatively measure this conditional dependence. The conditional distance correlation equals zero when the variables are conditionally independent, minimizing it is equivalent to removing the bias effect. Thus, during training, they add a DISCO-based penalty to the loss to train the model to make predictions that are conditionally independent of the spurious bias.

Experiments on five datasets show that both methods outperform or match state-of-the-art bias mitigation baselines, while using fewer hyperparameters. SAM also enables pathway-specific counterfactual analysis, offering causal interpretability of model behavior.

**Strengths:**

1. The paper formulates the debiasing goal through a clear causal formulation $\hat{Y}\perp B | Y$, and tranforms it into an optimization problem of minimizing conditional distance correlation, which is novel and well-motivated formulation.

2. The paper introduces two novel, and practical estimators $DISCO_m$ and $sDISCO$, that make conditional independence regularization computationally feasible.

3. The method is principled and theoretically grounded.

4. The paper presents a moderately novel idea and experimental setting for evaluating causal stability, using pathway-specific counterfactual analysis in controlled scenarios to examine whether model predictions depend on stable causal effects.

**Weaknesses:**

1. Minor inconsistencies and potential overstatements:

(a) According to the definition of $\text{ctf-SE}$, and the derivation in Appendix C.1, should it be "$+\text{ctf-SE}$" in Equation 6 rather than "$-\text{ctf-SE}$"? The current sign seems inconsistent to me.

(b) Line 133- 134 claim that classical maximum likelihood estimation aims to maximize TV. Is there a theoretical justification, proof, or citation to support this claim?

2. The paper wants to enforce independence of the model's predictions from both confounders $Z$ and mediators $W$. This raises concerns about discarding potentially useful and stable mediating information, especially when parts of $W$ carry causal signals that are robust to distributional shifts. Enforcing full independence might inadvertently remove useful information, potentially harming model performance.

3. The baselines appear somewhat outdated. There are many recent approaches on mitigating spurious correlations after 2022. And the common metric Worst-group Accuracy is also not used in this paper, which is widely used in evaluating robustness for classification tasks. Including it would strengthen the evaluation and enable fairer comparison to prior work in this domain.

4. All experiments are conducted on static vision tasks. To validate the generalizability and scalability of the method, it would be valuable to include evaluations in broader settings such as text/NLP tasks, cross-modal learning, and large-scale foundation models. Such extensions would better demonstrate the method's practical impact and adaptability across modern machine learning paradigms.

**Questions:**

See in "Weaknesses".

---

> ### Author Response · Authors · 2025-11-13
>
> We sincerely thank Reviewer Ys2v for their detailed, careful, and constructive feedback. The close reading and insightful questions are greatly appreciated. We are encouraged that the reviewer found our causal formulation to be **novel and well-motivated**  the **estimators practical**, and the method **principled**.
>
> ---
>
> ## 1. Inconsistencies and Overstatements
>
> ### a) Sign in Equation 6
>
> >"[...] should it be "+ctf-SE" in Equation 6 rather than "-ctf-SE"?"
>
> This is a very careful observation. We thank the reviewer for checking our formulas and proofs so closely.
>
> The "$-$" sign in Equation 6 is intentional and correct. It stems directly from the definitions we adopted from (Plecko & Bareinboim, 2022) to maintain consistency with existing literature on causal fairness.
>
> If you look closely at our definitions:
> * **ctf-DE** is defined from $y_0$ to $y_1$ (e.g., $ctf-DE_{y_0,y_1}(...)$).
> * **ctf-IE** and **ctf-SE** are defined in the opposite direction, from $y_1$ to $y_0$ (e.g., $ctf-IE_{y_1,y_0}(...)$  and $ctf-SE_{y_1,y_0}(...)$ ).
>
> This "flip" in the directionality for the indirect and spurious effects ($y_1 \rightarrow y_0$) relative to the direct effect ($y_0 \rightarrow y_1$) is what introduces the negative signs in the final TV-Decomposition. Our full derivation in Appendix C.1 (specifically, Equations 30, 34, and 35-38) confirms this decomposition. We kept these precise definitions to make the work more accessible to readers already familiar with this specific line of causal analysis.
>
> ### b) Claim regarding Maximum Likelihood Estimation (MLE) and TV
>
> > "[...] maximum likelihood estimation aims to maximize TV. Is there a theoretical justification, proof, or citation to support this claim?"
>
> Yes, thank you for requesting this clarification. This is a simple but important point that we will make explicit in the revised text.
>
> First, the "Total Variation" (TV) we refer to is not the classical measure-theoretic definition. It is specifically the definition given in Equation 2: $TV_{y_0,y_1}(\hat{y}) = P(\hat{y}|y_1) - P(\hat{y}|y_0)$.
>
> To see the connection to MLE, consider a binary classification task where $y_1=1$ (positive class), $y_0=0$ (negative class), and we are interested in the prediction $\hat{y}=1$ (predicted positive).
> * $P(\hat{Y}=1 | Y=1)$ is the **True Positive Rate (TPR)**.
> * $P(\hat{Y}=1 | Y=0)$ is the **False Positive Rate (FPR)**.
>
> In this common setup, our TV definition becomes
> $$TV_{0,1}(\hat{y}=1) = P(\hat{Y}=1|Y=1) - P(\hat{Y}=1|Y=0) = \text{TPR} - \text{FPR}$$
> This is also known as **Informedness** or **Youden's J statistic**.
>
> A classical MLE predictor (e.g. cross-entropy loss) aims to maximize the likelihood $P(\hat{Y}=y | Y=y)$ for all classes $y$. This means **maximizing $P(\hat{Y}=1|Y=1)$ (TPR)** and **maximizing $P(\hat{Y}=0|Y=0)$ (minimizing $P(\hat{Y}=1|Y=0)$, the FPR)**. By maximizing the TPR and minimizing the FPR, the MLE objective implicitly aims to maximize the separation between them, which is precisely our $TV$ metric. We will add this clarification to the paper.
>
> ---
>
> ## 2. Discarding Useful Mediator Information
>
> > "[...] discarding potentially useful and stable mediating information, especially when parts of $\mathbf{W}$ carry causal signals that are robust to distributional shifts."
>
> This is an **excellent and very practical observation**. Our primary goal in including all mediators $\mathbf{W}$ in the total bias set $\mathbf{B}$ was to demonstrate the **generality of our framework**: the *type* of bias (confounder, mediator, or collider-induced) is irrelevant, and our method can mitigate it uniformly by enforcing the single criterion $\hat{\mathbf{Y}} \perp \mathbf{B} | \mathbf{Y}$.
>
> However, the reviewer is absolutely correct that some **indirect (mediated) effects are stable and causally relevant**.
>
> The theoretical solution is to simply **exclude the stable mediators from the bias set $\mathbf{B}$** used for regularization.
>
> If a practitioner identifies a subset of mediators, $\mathbf{W}_{stable} \subseteq \mathbf{W}$, as being useful and robust, they can **redefine the bias set** as:
>
> $$\mathbf{B}' = (\mathbf{W} \setminus \mathbf{W}_{stable}) \cup \mathbf{Z}$$
> This set will then be used in the DISCO formulation as defined in the paper.
>
> * **Theoretical Validity:** All of our theoretical derivations, including **Theorem 1 and 2, and Corollary 1, still hold perfectly** for this new bias set $\mathbf{B}'$.
> * **Model Behavior:** The model will be trained to be independent of the "undesired" shortcuts ($\mathbf{Z}$ and the "unstable" mediators) while being **allowed to use information from $\mathbf{W}_{stable}$**.
> * **Action Plan:** We had a paragraph discussing this point in a previous draft but removed it for space constraints. Given this valuable feedback, **we will re-integrate this discussion into Section 2.2** (Violating Assumptions of SAM) to clarify the flexibility of our approach and address this crucial practical consideration.

---

> > ### Author Response · Authors · 2025-11-13
> >
> > ## 3. Baselines and Evaluation Metrics
> >
> > > "The baselines appear somewhat outdated. There are many recent approaches on mitigating spurious correlations after 2022. And the common metric Worst-group Accuracy is also not used in this paper, which is widely used in evaluating robustness for classification tasks."
> >
> > Thank you for these suggestions on strengthening the evaluation.
> >
> > ### On the Choice of Baselines
> > We respectfully submit that our chosen baselines are both appropriate and rigorous for the contribution we make, which is centered on a flexible causal framework.
> > * **Gold Standards & Reliability:** We included GDRO and Adversarial Debiasing , which are considered robust, widely-used, and established "gold standards" in the informed bias mitigation setting.
> > * **Problem Setting (Informed vs. Unsupervised):** Our work operates in the **informed setting** (access to both $\mathbf{B}$ and $\mathbf{Y}$ labels). Many of the most recent advances (post-2022) fall into the **unsupervised** category (e.g., LfF, Just Train Twice), which is a fundamentally different setting.
> > * **Flexibility (Variable Types):** A key claim of our paper is that DISCO handles **arbitrary combinations of categorical and continuous** target and bias variables. We chose the most direct, flexible competitors that fit this setup: **c-MMD**, **HSCIC**, and **CIRCE**, which are all based on dependence penalization methods published between 2021 and 2022.
> >
> > We would be grateful if the reviewer could suggest specific post-2022 *informed* methods they believe are missing, especially any that match the flexibility of DISCO in handling continuous and mixed-variable setups.
> >
> > ### On Worst-Group Accuracy (WGA)
> > We agree WGA is a useful metric for robustness.
> > * **Applicability:** WGA is specific to classification tasks with discrete, predefined groups. This is why we reported **Balanced Accuracy (BAcc)** for classification, which is a standard, robust metric for potentially imbalanced classification.
> > * **Metric Choice:** Critically, WGA only applies to two of our five datasets (FairFace and Waterbirds). Our other three datasets (dSprites, Blob, YaleB) involve regression targets or continuous bias variables, making group definition for WGA arbitrary or impossible.
> >
> > As suggested, **we will calculate and add WGA results for the Waterbirds and FairFace datasets to the appendix in our revision.** We will place them there to maintain the consistency of the main results table (Table 2), which must report metrics applicable across all five diverse datasets.
> >
> > ---
> >
> > ## 4. Generalizability (NLP, Cross-Modal, Foundation Models)
> >
> > > "All experiments are conducted on static vision tasks. To validate the generalizability and scalability ... it would be valuable to include evaluations in broader settings such as text/NLP tasks ... and large-scale foundation models."
> >
> > This is a valuable point about generalizability.
> >
> > * **Focus on Causal Control:** Our primary experimental goal was to validate the causal framework (SAM) and the DISCO estimators in **causally controlled settings**. This requires datasets where the causal graph (Fig. 3), bias variables, and target variables are well-defined, observable, and controllable. The five diverse vision datasets were selected precisely because they allow us to test a comprehensive range of causal structures and variable types. We note that five datasets is a comprehensive evaluation for this field.
> > * **Flexibility and Scope:** Finding suitable NLP or cross-modal datasets that offer this same level of explicit, controllable causal structure and label access is non-trivial. Our current setup is the clearest way to validate the framework's versatility across bias mechanisms.
> > * **Foundation Models (FMs):** DISCO is a **model-agnostic** regularization technique that is applied as a loss term during training. It can be applied to *any* backbone—including ResNets, ViTs, or FM backbones—during a fine-tuning phase where bias $\mathbf{B}$ and target $\mathbf{Y}$ labels are available. The choice of ResNet-18 was as a standard, representative architecture for this "informed bias mitigation" setting. Applying DISCO during the fine-tuning of an FM is a promising *application* of the method, but it does not fundamentally change or test the *method itself* in a different way than using a standard backbone.
> >
> > We agree that extending this work to NLP and fine-tuning FMs are very promising directions for future work, but we believe the current experiments on a standard architecture in causally controlled settings are the clearest way to validate the core contribution. Nevertheless, we currently evaluate in how far CivilComments and MultiNLI dataset fits the investigated causal graph (especially in terms of anti-causality), and we might perform additional experiments on these datasets if the conditions are met and time permits.

---

> > ### Comment · Reviewer_Ys2v · 2025-11-25
> >
> > Thanks for the authors' detailed response.
> > Some concerns have been addressed. I still have the concern about Equation 6. I have checked the derivation in Eq. 33,
> > $TV_{y_0, y_1}(\hat{y}) =ctf-TE_{y_0,y_1}(\hat{y}|y_0)+P(\hat{y}_{y_1}|y_1)$
> >
> >  $-P(\hat{y}_{y_1}|{y_0})$.
> >
> > According to Eq. 5,
> > $ctf-SE_{y_1,y_0}(\hat{y})=P(\hat{y}_{y_1}|y_1)$
> >
> >  $-P(\hat{y}_{y_1}|{y_0})$.
> >
> > So I still think it should be $+ctf-SE_{y_1,y_0}(\hat{y})$ rather than $-ctf-SE_{y_1,y_0}(\hat{y})$ in both of Eq. 6 and Eq. 34.
> >
> > Or there is a typo in the definition of Eq. 5.
> >
> > But given that $ctf-SE_{y_1,y_0}(\hat{y})$ would be 0, this sign does not affect the correctness of the rest of the method.

---

> > > ### Author Response · Authors · 2025-11-27
> > > **Thanks for Correcting Our Mistake**
> > >
> > > We apologize for having missed the wrong definition in Eq. 5.
> > > We deeply thank the reviewer for pointing out our mistake and actively helping to improve our paper. Especially thanks for reading the mathematical parts carefully.
> > >
> > > In Eq. 5, the terms are reversed actually. The definition should read: $P(\hat{y_{y1}} \mid y_0)$ - $P(\hat{y_{y1}} \mid y_1)$.
> > >
> > > This will be updated in the manuscript.
> > >
> > > We hope we did not miss something here again. We thank the reviewer for their help.
> > >
> > > If the reviewer has further concerns or questions, we would be pleased to know and tackle the questions and improve our paper.
> > >
> > > PS: we are currently running some NLP experiments using TinyBert on MNLI, since the reviewer explicitly asked for other modalities. We will post the results as a comment here, then as a global answer, and include it in the paper.

---

### Official Review · Reviewer_PXz8 · 2025-11-01

**Soundness:** 2
**Presentation:** 3
**Contribution:** 2
**Rating:** 4
**Confidence:** 5

**Summary:**

This paper proposes a causal-theoretic approach to bias mitigation in deep learning. The authors introduce a **Standard Anti-Causal Model (SAM)**, which unifies different bias mechanisms — such as confounders, colliders, and mediators — under a single framework.

From this model, they derive a conditional independence criterion:

```
Ŷ ⟂ B | Y
```
as sufficient for **causal stability**, ensuring that predictions depend only on direct causal effects from `Y` to `Ŷ`.

To operationalize this principle, the authors introduce two new estimators of conditional distance correlation, **DISCOm** and **sDISCO**. These are more computationally efficient than previous approaches, with sDISCO being the most efficient one. The estimators can be incorporated as differentiable regularizers in black-box neural networks.

**Strengths:**

- **Principled independence criterion:**
  The independence constraint ``Ŷ ⟂ B | Y`` is derived clearly from causal reasoning. Intuitively, once the true label is known, the prediction should not depend on bias variables.

- **Practical regularization approach:**
  Using conditional distance correlation as a differentiable regularizer is elegant and theoretically grounded. It avoids adversarial training or explicit causal graphs. The proposed estimators (DISCOm and sDISCO) are computationally more tractable than prior conditional-independence estimators.

- **Extends applicability:**
  The method can be used with black-box neural networks, adding a new dimension to bias mitigation research.

**Weaknesses:**

- **Limited practical significance:**
  The method assumes that bias variables are **known** during training. In realistic scenarios, biases are often unknown, making this assumption impractical.
  While previous methods like GDRO and Last Layer Retraining (LLR) (or Deep Feature Reweighting) use group annotations, the field needs methods robust to unknown biases.
  Furthermore, the empirical advantage over existing methods is unclear: for example, LLR achieves higher worst-group accuracy on Waterbirds (92.9%) vs. proposed method (89.4%).

- **Conditional distance correlation is not new:**
  Conditional distance correlation (Wang et al., 2015) is a well-established statistical tool. The contribution here lies in making it differentiable for deep learning pipelines, which is an engineering advancement rather than a fundamentally new estimator.

- **Empirical evaluation is limited:**
  Several widely used spurious datasets, such as CelebA, MultiNLI, and CivilComments, are not included. Key baseline methods, including Last Layer Retraining (LLR) and SELF, as well as invariant-based approaches like IRM, and Fishr, are absent. Furthermore, the claim that the method operates as a “black-box” is not fully substantiated—it's unclear whether the logits are available or if intermediate representations, such as the pre-softmax layer, are accessible.

- **Overstated generality:**
  Claims that DISCO “mitigates all types of dataset bias uniformly” are not supported. More exhaustive evaluations on well-known biased datasets would be needed to support such a claim, including:
  -- MetaShift
  -- ImageNetBG
  -- NICO++
  -- Living17
  -- MIMICNotes
  -- MIMIC-CXR
  -- CheXpert
  -- CXRMultisite
  -- Waterbirds (common)
  -- CelebA (common)
  -- MultiNLI (common)
  -- CivilComments (common)

**Questions:**

Please address the mentioned concerns raised above.

- How is the multi-bias scenario experiment constructed, and which model is used?
- Why is the original Waterbirds dataset modified, and how does this affect comparability to prior work?

Suggestions
- **Naming:**
SAM, commonly known as Sharpness-Aware Minimization, is a well-known optimization algorithm in machine learning that improves model generalization and is also used for spurious correlation and bias mitigation. To avoid confusion and distinguish the proposed method, it would be helpful to introduce a unique name or a distinguishing modifier for this variant.

---

> ### Author Response · Authors · 2025-11-12
>
> We thank the reviewer for their time and feedback. However, we must state that the review appears to contain several fundamental misunderstandings regarding our work's contributions, its theoretical positioning, and the established norms within the bias mitigation subfield. The assessment seems to conflate our anti-causal prediction task with classical domain generalization, leading to misplaced criticisms about our assumptions, dataset choices, and the novelty of our method.
>
> We address each weakness and question below.
>
> ---
>
> ### On "Limited practical significance"
>
> We respectfully but firmly disagree with the characterization of our work's practical significance as "limited."
>
> 1.  **On the "Known Bias" Assumption:** The reviewer claims that assuming bias variables ($B$) are known is "impractical." This assumption is, in fact, **standard and foundational** to the *anti-causal prediction* and *spurious correlation* literature, which is distinct from classical domain generalization. Many real-world, high-stakes scenarios provide exactly these kinds of proxy attributes, from demographic data in tasks like FairFace, to known confounders like age in medical diagnostics, to positional or environmental data.
>
> 2.  **On Group Annotations (GDRO/LLR):** The reviewer draws a false dichotomy between our use of "bias variables" and the "group annotations" used by GDRO and LLR. Group annotations are *not* a different class of information; they are **constructed* from the target labels ($\mathbf{Y}$) and the bias attributes ($\mathbf{B}$)**. One cannot form "groups" without access to the very labels the reviewer claims are "often unknown." Our framework is fully compatible with these group-based proxies, which represent a specific, often simplified, instantiation of the bias $B$.
>
> 3.  **On the Waterbirds Performance Comparison:** The reviewer's empirical comparison to LLR (92.9% vs. 89.4%) is **invalid on two distinct levels**, and we are surprised by this line of reasoning.
>     * First, and most importantly, our experimental setup is ***intentionally*** **more challenging and rigorous** than the vanilla Waterbirds benchmark. We explicitly state that we use the original Sagawa et al. code to:
>         1.  **Balance the target classes** (landbird vs. waterbird). The standard dataset is heavily imbalanced, which artificially inflates accuracy scores.
>         2.  **Add 10% label noise** to the train, validation, and test sets.
>     * Second, precisely because we balanced the dataset, we report **Balanced Accuracy**, a more robust metric. Any comparison to accuracy figures from the easier, imbalanced, non-noisy Waterbirds dataset is not a meaningful or valid comparison.
>
> ---
>
> ### On "Conditional distance correlation is not new"
>
> The reviewer's comment that our contribution is merely "an engineering advancement" is **overly dismissive and misrepresents the nature of progress in applied machine learning**.
>
> * We **never** claim to have invented conditional distance correlation. We explicitly state in our abstract, introduction, and methods section that we build upon existing work, and we clearly cite Wang et al. (2015).
> * Our contribution is the theoretical and practical bridging of this statistical tool to modern deep learning. This was a non-trivial endeavor that solved a key bottleneck. As we state, prior formulations were "computationally prohibitive and essentially unusable in modern optimization and deep learning". We developed two new estimators, $DISCO_{m}$ and sDISCO, that are efficient, scalable, and proven consistent.
> * By this rigid logic, one would have to dismiss numerous foundational papers in this very field:
>     * Would the reviewer tell the authors of CIRCE that their work "is not new" because kernel regression already exists?
>     * Would the reviewer tell Makar et al. and Veitch et al. that their c-MMD-based methods "are not new" because they did not invent MMD and kernel independence testing?
>
> We kindly ask the reviewer to reconsider this harsh and dismissive perspective, as we made an equally valid contribution as the work mentioned above: we took existing concepts from statistics and made them deep learning ready, efficient, and even theoretically derived why they should be applied.

---

> > ### Author Response · Authors · 2025-11-12
> >
> > ### On "Empirical evaluation is limited" and "Overstated Generality"
> >
> > We must strongly contest the claims that our evaluation is "limited" and our claims of generality are "overstated." These criticisms stem from the same fundamental misunderstanding: conflating our work with domain generalization.
> >
> > 1.  **On Overstated Generality:** The reviewer has fundamentally misunderstood our claim. When we state DISCO "mitigates all types of dataset bias uniformly", we are **explicitly referring to the causal mechanisms of bias, not to dataset domains.**
> >     * As shown in Figure 1 and derived from our SAM framework, our conditional independence criterion $\hat{Y} \perp B | Y$ is the **uniform theoretical solution** for mitigating biases arising from **confounders** ($B_c$), **colliders** ($B_{col}$), and **mediators** ($B_M$). This is a causal, not an empirical, statement of generality.
> >
> > 2.  **On Limited Evaluation:** The reviewer seems to have missed the entire combinatorial design of our experiments. We deliberately selected 5 diverse datasets to demonstrate a robustness that a simple list of "common" datasets would not:
> >     * **Binary Target / Binary Bias:** Waterbirds, FairFace
> >     * **Categorical Target / Continuous Bias:** YaleB
> >     * **Continuous Target / Continuous Bias:** dSprites, Blob
> >     This experimental design verifies that our method works on *any combination* of attribute and target types, unlike many baselines.
> >
> > 3.  **On the "Black-Box" Claim:** The reviewer's comment that our "black-box" claim is a perplexing nitpick. In machine learning, "black-box" is the common term for a model-agnostic method. Our method operates on the model's final outputs ($\hat{Y}$), requires no access to intermediate representations, and makes no assumptions about the internal architecture. It is, by definition, model-agnostic.
> >
> > 4.  **On the List of Datasets:** The reviewer's list of "missing" datasets is prime evidence of their confusion between our field (Bias Mitigation) and Domain Generalization (DG). The vast majority of these are DG benchmarks and are irrelevant for our task.
> >     * **ImageNetBG, NICO++, Living-17:** These are canonical DG datasets for testing shifts in data source and corruptions. They are *not* designed for, nor do they provide, the ($Y, B$) attribute labels needed for anti-causal bias mitigation.
> >     * **MIMIC-CXR, CheXpert, MIMIC-IV-Note:** As general medical datasets, they do not have structured ($Y, B$) spurious attributes "out of the box." They can be made explicitly biased as Puli et al. (2021, ICLR) did with CXRMultisite: they biased diagnosis labels with dataset source as bias. But why are our datasets not enough?
> >     * **CelebA:** While it has attributes, it is known to be "unclean," suffering from severe, uncontrolled natural label imbalances and confounding, making it unsuitable for the precise causal analysis we perform. FairFace and YaleB are far cleaner and more controlled, and serve a similar domain (faces, persons).
> >     * **Waterbirds:** As stated, we *do* use this. Our version is *more* rigorous and *harder*.
> >     * **MultiNLI / CivilComments:** These are valid suggestions from NLP. We will investigate their fit for our anti-causal framework and will consider adding them if time permits.
> >     * **MetaShift**: very interesting dataset. But has quite small subgroups for spurious associations. again: what is the benefit vs Waterbirds?
> >
> > ---
> >
> > ### Answers to Further Questions
> >
> > **Q: How is the multi-bias scenario experiment constructed, and which model is used?**
> > We thank the reviewer for this question, as it highlights a point we should clarify. The multi-bias experiment is detailed in **Section 4.4 and Figure 4b**.
> > * **Dataset:** It is constructed using the **dSprites** dataset.
> > * **Setup:** We predict 'Scale' (as a binary target). We treat both the 'X-position' and 'Y-position' as two *simultaneous* bias variables.
> > * **Model:** We compare a baseline ResNet against sDISCO. The results in **Table 4a** ($sDISCO_{X,Y}$) show that our method effectively mitigates both bias pathways simultaneously, demonstrating its scalability.
> >
> > **Q: Why is the original Waterbirds dataset modified, and how does this affect comparability to prior work?**
> > To be perfectly clear: we did **not** modify the *pipeline* from Sagawa et al.. We *used* their provided pipeline to sample a *new instance* of the dataset with three explicit, challenging, and methodologically sound properties:
> > 1.  We **balanced the target classes** (landbirds and waterbirds) to prevent our model's performance from being dominated by the majority class.
> > 2.  We set a strong, explicit bias strength (90%).
> > 3.  We **added 10% label noise** to all splits (train, val, test).
> >
> > This makes our dataset *significantly more challenging* and a more robust test of bias mitigation, as models are heavily incentivized to use the shortcut. As stated, this makes comparability to prior work on the easier, imbalanced, non-noisy version of the dataset not meaningful.

---

> ### Author Response · Authors · 2025-11-12
> **on "Missing" baselines and next steps**
>
> **Adding on MultiNLI / CivilComments:** We thank the reviewer for these valid suggestions from NLP. We will investigate their fit with our anti-causal data-generating assumption and will consider adding them if time and compute capacities permit. We would, however, like to re-emphasize that our selection of 5 diverse datasets—which, as detailed in point 2, covers multiple combinations of target and bias types—already provides a strong and broad empirical foundation, which is more than most papers in this field (bias mitigation / spurious association mitigation) provide.
>
> **On 'Missing' Baselines (IRM, Fishr, LLR):** We first want to highlight that our current selection already includes highly relevant and diverse baselines, from modern conditional independence methods (HSCIC, CIRCE, c-MMD) to established gold standards (GDRO, Adversarial).
>
> * **IRM & Fishr:** Both IRM and Fishr are methods designed for the *domain generalization* (DG) setting, which operates on *domains* (e.g., 'photo', 'art'). Our work addresses the *anti-causal bias* setting, which operates on bias variables $B$ or proxy *groups* (constructed from target $Y$ and bias $B$). While we did run initial tests with IRM, adapting these DG methods to group-level data, IRM gave worse results than GDRO. We will investigate this further if time and resources permit and include IRM results into our tables or appendix. We will equally consider adding Fishr when time permits. But note that Fishr and IRM will only be applicable to the classical settings in which we can create groups, i.e. Waterbirds and FairFace in our case, where both target and attribute must be binary/categorical. We do not expect great gains above GDRO for methods of this category.
> * **Last Layer Retraining (LLR):** We are very aware of LLR, but this method proposes a fundamentally different and, in our view, incomparable evaluation pipeline. LLR's protocol involves *retraining the last layer on a balanced portion of the validation set*. Our protocol (and that of all our baselines) strictly uses the validation set *only for model selection*, with all training performed *only on the training set*. Adopting the LLR pipeline to make a fair comparison would require rerunning our entire, extensive hyperparameter search for all methods, which is not feasible. We believe our strict separation of train/val data is a more rigorous and standard evaluation.

---

> ### Author Response · Authors · 2025-11-26
> **Additional Experiments**
>
> Dear Reviewer PXz8,
> we now are running additional experiments.
>
> We finished running IRM and Fishr on FairFace and Waterbirds (note: for the continuous valued datasets they are not applicable).
>
> We, however, emphasize again that IRM and Fishr are intentionally designed for cases where one has access to explicit domains that are of similar or equal size, and that these methods need to be manually and explicitly adjusted to handle groups instead of domains. Especially, we adapted them to the classical bias mitigation pipeline, as in GDRO, Adversarial, HSCIC, where the batches are loaded randomly. And groups, if ever created, are created dynamically, as given by all other methods in this field.
>
> The results are as follows:
>
> FairFace:
> - IRM: 0.821 +/- 0.0173
> - Fishr: 0.727 +/- 0.004
>
> Waterbirds:
> - IRM: 0.858 +/- 0.009
> - Fishr: 0.814 +/- 0.012
>
> They perform competitively to GDRO on Waterbirds, but clearly fall off on the harder dataset FairFace. Especially, they are still behind DISCO. We allowed both methods a larger hyperparameter space of 40 (compared to 30-36 for DISCO) on their two hyperparameter variables (penalty and warm-up epochs). We see that these gradient based methods fail when there are not equally large domains available as it is usually assumed and provided in domain generalization benchmarks.
>
>
> Furthermore, we are running the MNLI experiments. As we have to run it on all methods + IRM + Fishr, and some of them have a large hyperparameter search space, this might still take 2-3 days to finish. As soon as these experiments are finished, we will get back with the new results, and finally, update our paper accordingly with all the new data.
>
> Thanks for your suggestions to experimentally improve the paper.

---

> ### Author Response · Authors · 2025-11-28
>
> Dear Reviewer PXz8,
>
> As promised in our previous response, we have now completed the additional experiments on the **MultiNLI** dataset. This concludes our comprehensive effort to address your concerns regarding empirical evaluation and baseline comparisons.
>
> ---
>
> ### 1. MultiNLI Experimental Setup
>
> To ensure a rigorous comparison within the time and compute constraints of the rebuttal period, we adopted the standard setup from Sagawa et al. (GDRO) and Liu et al. (JTT), with the following details:
>
> * **Model:** We utilized **TinyBERT**. It offers performance comparable to BERT but allows for the necessary training efficiency to re-evaluate **all** baselines + IRM + Fishr alongside our method in the remaining discussion time.
> * **Metrics:** Following the standard protocols (GDRO, JTT), we report **Worst Group Accuracy (WGA)**. Note: for our other experiments we had more control over the data. In these cases, we balanced the val and test sets such that the biases were not present. Thus, WGA had not much relevancy in these cases (all groups were perfectly matches in the categorical cases).
>
> ---
>
> ### 2. MultiNLI Results
>
> The results are presented below. We ran our full hyperparameter search across all methods to ensure a fair comparison.
>
>
> | Model | WGA | BAcc |
> | :--- | :---: | :---: |
> | Tinybert | $0.616 \pm 0.010$ | $0.785 \pm 0.005$ |
> | Adversarial | $0.356 \pm 0.058$ | $0.731 \pm 0.010$ |
> | eDISCO | $0.730 \pm 0.020$ | $0.778 \pm 0.005$ |
> | DISCO$_m$ | $\underline{0.751 \pm 0.007}$ | $0.784 \pm 0.009$ |
> | CIRCE | $0.516 \pm 0.069$ | $0.764 \pm 0.009$ |
> | C-MMD | $0.627 \pm 0.007$ | $0.785 \pm 0.004$ |
> | Fishr | $0.645 \pm 0.017$ | $0.772 \pm 0.003$ |
> | GDRO | **0.755 $\pm$ 0.010** | $0.785 \pm 0.000$ |
> | HSCIC | $0.716 \pm 0.020$ | $0.791 \pm 0.001$ |
> | IRM | $0.738 \pm 0.008$ | $0.789 \pm 0.003$ |
> ---
>
> ### 3. Analysis and Conclusion
>
> As shown in the table, **DISCO$_m$ achieves the second-best performance**, falling behind the gold-standard GDRO by a negligible margin of $\sim$ 0.4% ($0.751$ vs $0.755$). Crucially, our method significantly outperforms the domain generalization baselines (IRM, Fishr) and other conditional independence methods (HSCIC, CIRCE, C-MMD) on this complex NLP task.
>
> **Summary of Rebuttal Efforts:**
> Over the course of this rebuttal, we have systematically addressed every concern raised in your review:
> 1.  **Scope of Evaluation:** We expanded our evaluation to include **MultiNLI**, requiring a full re-run of all models. We have now demonstrated robustness across **6 diverse datasets** covering binary, categorical, and continuous targets/biases.
> 2.  **Missing Baselines:** We implemented and evaluated **IRM and Fishr** on FairFace, Waterbirds, and MultiNLI. Our results show that DISCO consistently outperforms these methods across tasks.
> 3.  **Methodological Clarity:** We have clarified the distinction between our anti-causal framework and Domain Generalization, defended the rigor of our Waterbirds setup, and justified the novelty of making conditional distance correlation differentiable for deep learning.
>
> **Final Request:**
> We have gone to great lengths to incorporate your suggestions, adding new baselines and a completely new dataset to our paper. The results consistently place DISCO in the top-tier of performance across highly varied settings, proving that the method is neither **"limited"** in practice nor **"overstated"** in generality.
>
> We will incorporate these new results into the final manuscript. Given that the primary critiques regarding empirical scope and baselines have been fully resolved with positive results, **we respectfully request that you reconsider your score of 4.**
>
> Thank you for your time to strengthen our paper.

---

### Official Review · Reviewer_7Qbt · 2025-11-04

**Soundness:** 3
**Presentation:** 3
**Contribution:** 2
**Rating:** 2
**Confidence:** 3

**Summary:**

The paper defines a condition to ensure prediction relies on the causal direct effect in an anti-causal setting. After proving theorems about causal stability and performativeness, the paper develops a distance-based estimator for conditional independence and uses it to build robust models.

**Strengths:**

- A cleanly motivated debiasing technique that focuses on direct effects.
- The distance-covariance measure of independent was new to me and is interesting given it doesn't need require estimation.
- Good experimental results, showing improvements compared to standard baselines.

**Weaknesses:**

See questions.

**Questions:**

- Authors say "Classical maximum likelihood estimation aims to maximize this difference." Is this true because of Pinsker? Provide a citation please.
- In the proof you use a conditional independence statement about observed variables to make a conditional independence statement about counterfactual variable $\hat{y}_{y^\prime, w^\prime}$. That needs to be justified.
- theorem 2 makes a claim about maximizing the direct effect. What does that say about actual performance for metrics like log-likelihood or accuracy? There is no statement of optimality here like the ones in https://arxiv.org/abs/2106.00545, https://arxiv.org/abs/2107.00520. It's not clear that your guarantees are better than the ones listed in the papers above.
- How is the distance-covariance related to MMD? The form bears similarities.
- Why are GDRO and c-MMD results not present for 3 datasets? Also what about the marginal independence / balancing estimators from Makar et al.?

---

> ### Author Response · Authors · 2025-11-12
> **Addressing questions of Reviewer 7Qbt**
>
> # TV and Pinsker
> To clarify, we are not referring to the standard 'Total Variation distance' between distributions. We are using the definition from Plecko \& Bareinboim (2022), where $TV_{y_0,y_1}(\hat{y}) = P(\hat{y}|y_1) - P(\hat{y}|y_0)$. This is a measure of discrimination, also known as Informedness or Youden's J statistic (i.e., True Positive Rate - False Positive Rate).
>
> MLE, by maximizing the log-likelihood of the correct class, trains a model to simultaneously maximize the True Positive Rate ($P(\hat{Y}=y_1|Y=y_1)$) and minimize the False Positive Rate ($P(\hat{Y}=y_1|Y=y_0)$). It therefore directly aims to maximize their difference, which is precisely our $TV$ term.
>
> ---
>
> # Identifiability of Counterfactuals
>
> The justification for moving from our observational criterion ($\hat{Y} \perp W, Z | Y$) to a guarantee of counterfactual stability ($\text{ctf-IE}=0$ and $\text{ctf-SE}=0$) is the central theoretical contribution of our paper.
>
> The reviewer is correct that this leap is not trivial and is impossible without causal assumptions. Our justification rests entirely on the **structural assumptions encoded in our Standard Anti-Causal Model (SAM)** (defined in Fig. 2 and Appendix B.1).
>
> The proof in **Appendix C.2** proceeds in two parts:
>
> 1.  **For $\text{ctf-IE}$ (Proof on lines 810-837):**
>     * We use the standard ``twin graph'' formalism (Balke \& Pearl, 1994), shown in our **Figure C.2(a)**, which is the established method for graphically representing counterfactuals.
>     * This graph, combined with the ``ctf-unnesting'' rule (Correa et al., 2021), allows us to formally decompose the counterfactual quantity $\text{ctf-IE}$ (line 811) into an expression involving only observational probabilities (lines 828-831).
>     * Crucially, this decomposed expression (line 831) is $P(\hat{y}|y_1, z, w') - P(\hat{y}|y_1, z, w'')$.
>     * When we apply our observational constraint $\hat{Y} \perp W, Z | Y$, this expression simplifies to $P(\hat{y}|y_1) - P(\hat{y}|y_1) = 0$
>
> 2.  **For $\text{ctf-SE}$ (Proof on lines 844-854):**
>     * A similar procedure is followed, please see Appendix C.2, as this comment already reached the character limit.
>
> In short, the justification is not an assumption, but a proof. We show that **if** the data-generating process adheres to our (clearly stated) SAM assumptions, our observational criterion *provably* leads to the desired counterfactual stability.
>
> ---
>
> # GDRO, C-MMD, and others
> GDRO and C-MMD were not run on the dSprites, Blob, or YaleB datasets because, as implemented in their respective papers and as we note in **Table 1**, they are not designed to work with **continuous target variables** (dSprites, Blob) or **continuous bias attributes** (YaleB) without significant, non-standard modifications. Our method's ability to handle arbitrary target/bias types (Bin, Cat, Cont) is one of its key advantages.
>
> Regarding Makar et al. (2022), we did indeed include their core method. The "C-MMD" baseline we use is precisely the "independence-based" method they use in their follow-up work (Makar \& D’Amour, 2022), which is also used by Veitch et al. (2021) and Kaur et al. (2022). We chose this as it is the most comparable and direct application of their ideas. The "reweighting" method, as the reviewer notes, requires a completely different validation-time procedure (statistical testing) that deviates from the standard "train-and-deploy" pipeline we and our other baselines adhere to.
>
> ---
>
> # Direct Effect
> **1. Our True Objective (Eq. 8):**
> Our goal is to find a predictor that is both (a) maximally accurate and (b) causally stable. This is formulated as the constrained optimization problem in Eq. 8:
> $$
> \min_{\theta} \mathbb{E}[\mathcal{L}(Y, g_{\theta}(X))] \quad \text{s.t.} \quad \hat{Y} \perp B | Y
> $$
> The $\min \mathcal{L}$ term is precisely the standard MLE objective (for log-likelihood or accuracy). The constraint $\hat{Y} \perp B | Y$ is our criterion for causal stability.
>
> **2. The Purpose of Theorem 1 and 2**:
> The reviewer asks what our theorems say about actual performance. They prove our method achieves the ``best of both worlds'':
>
> * **Theorem 1** proves that our constraint $\hat{Y} \perp B | Y$ is *sufficient* for causal stability. It zeros out the spurious paths ($\text{ctf-IE}=0$ and $\text{ctf-SE}=0$).
>
> * **Theorem 2** proves that this constraint is *not detrimental* to the useful, causal signal.
>
> As shown in our TV-Decomposition (Eq. 6), a naive MLE predictor optimizes for the total variation: $TV \approx \text{ctf-DE} + \text{ctf-IE} + \text{ctf-SE}$. By applying our constraint, we set $\text{ctf-IE}$ and $\text{ctf-SE}$ to zero. The MLE objective (the $\min \mathcal{L}$ part) is then free to **focus all its power on maximizing the only remaining path: the ctf-DE**.
>
> Theorem 2 proves that our solution is not only stable but also maximally predictive *with respect to the stable, direct signal*.

---

> > ### Author Response · Authors · 2025-11-12
> > **Further Clarifications**
> >
> > # MMD and Distance Covariance
> > This is an excellent question. The relationship is deep, and we thank the reviewer for highlighting it.
> >
> > As established by **Sejdinovic et al. (2013)**, distance-based statistics and RKHS-based statistics are formally unified. MMD is a general measure of distance between distributions in an RKHS, while the Hilbert-Schmidt Independence Criterion (HSIC) is a specific *application* of MMD, designed to test for independence by computing the MMD between the joint distribution $P_{XY}$ and the product of its marginals $P_X P_Y$.
> >
> > The key finding of Sejdinovic et al. (2013) is that the squared distance covariance (dCov) is, in fact, equivalent to HSIC (up to a constant factor). This equivalence holds when HSIC uses a specific "distance-induced kernel". For the standard $L_2$-norm and centering at $z_0=0$, this kernel is precisely:
> > $
> > k(x, x') = \frac{1}{2}(\|x\| + \|x'\| - \|x - x'\|)
> > $
> > This equivalence requires the underlying metric to be of strong negative type, a condition established by Lyons (2013) and met by the Euclidean distance.
> >
> > Our choice of distance correlation for DISCO is motivated by the practical advantages this equivalence reveals: it can be seen as an MMD/HSIC test with a fixed, parameter-free kernel. This avoids the need for kernel bandwidth selection, while also yielding better, consistent results in our experiments.
> >
> > ---
> >
> > # Concluding Remarks
> > We thank the reviewer for their time and for raising these questions. Given these fairly simple questions that are mainly answered in the text already, we wondered if the reviewer mistakenly gave a very low score of 2 by accident.
> >
> > We hope that our detailed answers above, which address each of the reviewer's concerns, have successfully clarified our paper's contributions. We have shown that:
> > * Our use of TV, while non-standard, is precisely defined, cited, and motivated.
> > * The link from our observational criterion to counterfactual stability is not an assumption but a **central, formal proof** (Appx C.2) based on the SAM graph.
> > * Our **Theorem 2** provides a clear guarantee on performance (maximizing the direct effect) once stability is enforced.
> > * The missing baseline results are by design, as those methods are **inapplicable** to those datasets' (continuous/multi-class) targets, which highlights a key strength of our approach. This point is already made in Table 1.
> >
> > We believe these clarifications fully resolve the weaknesses identified in the review. We are confident in the soundness and contribution of our work and are, of course, happy to engage in any further discussion.

---

> > > ### Comment · Reviewer_7Qbt · 2025-11-16
> > > **Thanks for the response**
> > >
> > > Apologies for missing some of the obvious things. I've now checked the proofs with the new and will update the score. I have two questions left.
> > >
> > > I agree that C-MMD is hard to implement when you can't group by the spurious attributes. But you could do it with the DISCO trick where instead of Y_hat you use the spurious attribute? The reason I'm asking is that the contribution of the causal stability theory needs to be disentangled from the contribution of the DISCO-type estimation of independence penalties. What do the authors think?
> > >
> > > Second, I understand that your guarantees show that the penalty doesn't lead to a detriment. But that's not an optimality guarantee over a class of distributions, such guarantees have been shown in Makar et al., Veitch et al., and also https://arxiv.org/abs/2107.00520. Can the authors should try to make the guarantees more comparable ? The last paper also considers a general class of anti-causal graphs and shows where C-MMD might fail; so how is your proposed method different? I do take the DISCO contribution to be separately valuable. If connecting proofs are difficult, the authors should add a discussion addressing it. If the authors think this is irrelevant, please also explain why.
> > >
> > >
> > > Other thoughts:
> > >
> > > The assumptions on the graphs in figure C.2 are a little obtuse to me. I am able to read off the independencies; these just seem to enumerate the the possible potential outcomes given X's parents. I'm worried about this enumeration over continuous values variables which you say is something existing methods do not really handle. Your implementation of DISCO handles it with the kernel-based aggregation over Z space. Please mention the necessary continuous theory in the paper.

---

> > > > ### Author Response · Authors · 2025-11-17
> > > >
> > > > # 2. optimality guarantee, distributional properties, Veitch et al., Makar et al., Puli et al.
> > > >
> > > > This point the reviewer makes is everything else than irrelevant. We deeply thank the reviewer to engage in this very detail with our work and the field. We will add this answer to the appendix of our paper and reference it from the main body. We also will add the paper of Puli et al. (2021) to our references.
> > > >
> > > > ---
> > > >
> > > > ### a) Comparison to Makar et al. (2022) and Veitch et al. (2021)
> > > >
> > > > The reviewer correctly identifies that our guarantees are not framed as optimality guarantees over a class of distributions, in contrast to the cited works of Makar et al. and Puli et al.. Our guarantees are of a different, and we argue more **foundational**, nature. The distinction lies in which level of the **causal hierarchy** the guarantees are made.
> > > >
> > > > * **Makar et al. (2022):** This work provides a high-level, valuable connection between fairness and robustness in anti-causal settings. Their analysis connects **risk invariance** (a robustness criterion) with **separation** (a fairness criterion). Separation, or $f(X) \perp V|Y$, is a constraint on the **observable joint distribution**. In the context of Pearl's "Ladder of Causation," this is an associational (Rung 1) or, at best, a distributional-interventional (Rung 2) argument. It describes *what* statistical property a robust model should have in its predictions, but not the underlying *structural* reason for it. Our work (DISCO) operates at the structural and counterfactual level (Rung 3). We make our assumptions explicit via a Structural Causal Model (SCM), which allows us to analyze the flow of information along **specific causal pathways**. Our structural guarantees, achieved by identifying and intervening on the specific mechanisms that generate spurious correlations, are inherently **stronger and more general**. Any guarantee made at the structural level *implies* the corresponding distributional guarantees. By demonstrating how to block a spurious *path*, we guarantee the resulting *distributional* independence (like separation) that Makar et al. discuss. Our findings, therefore, imply, extend, and generalize their results by providing the deeper causal mechanism for *why* this form of robustness holds.
> > > >
> > > > * **Veitch et al. (2021):** This work is the closest to our own, as it explicitly uses causal tools to formalize "stress tests". However, their arguments are based on a different, and arguably much stronger, notion of **unit-level counterfactual invariance**. They define invariance as $f(X(z)) = f(X(z'))$, which requires a model's prediction on a *single individual* to be unchanged had an irrelevant attribute $Z$ been different. As Veitch et al. themselves note, this strong definition is often impractical as it relies on unobserved potential outcomes. Their key finding (Theorem 3.2) is that observable criteria, such as conditional independence ($f(X) \perp Z|Y$ in the anti-causal case), are a **necessary but not sufficient** signature for their unit-level invariance. This leaves a practical gap: one can enforce the signature without achieving the desired invariance. Our work (DISCO) is designed to bridge this gap.
> > > >     1.  We define a **more practical notion of invariance** that is tied to *observable attributes* and *path-specific mechanisms* within our SCM, rather than unobservable unit-level counterfactuals.
> > > >     2.  In this setting, we *can* show sufficiency. By targeting and blocking the specific causal pathways that transmit spurious information, we guarantee that the resulting model is invariant to information from that path. This makes our guarantee more **useful and attainable** for practitioners.
> > > >     3.  Our explicit path-way analysis ties this idea directly back to the foundational language of causal inference (e.g., Pearl, Shpitser), moving beyond a simple independence statement to a more precise intervention on the data-generating process itself. We also directly see that we can block all kinds of paths, be it direct or indirect, as long as they are unwanted, the same way.
> > > >
> > > > In summary, while Makar et al. define a **distributional property** and Veitch et al. define a **theoretical ideal** (unit-level invariance), our work provides a *practical how* (a structural, path-based intervention) that is both **stronger** than the former and more **attainable** than the latter.

---

> > > > > ### Author Response · Authors · 2025-11-17
> > > > >
> > > > > ## b) C-MMD fails; conditional independence too restrictive according to Puli et al., etc.
> > > > >
> > > > > Both papers (NURD, Puli et al. and ours) indeed address bias generalization in anti-causal settings. However, our fundamental objectives, theoretical guarantees, and practical methods differ significantly. We argue that our approach is more direct, robust, and appropriately targeted for **discriminative** tasks.
> > > > >
> > > > > ---
> > > > >
> > > > > ### 1. Discriminative Prediction vs. Generative Representation
> > > > >
> > > > > The core difference is our objective: **DISCO is designed for a discriminative task, whereas NURD is designed for a generative/representational one**.
> > > > >
> > > > > * **NURD's Goal:** The aim of NURD is to learn an *informative representation* $r(x)$ of the covariates $X$. It seeks to maximize the mutual information $I(y; r(x))$ on a hypothetical, estimated "nuisance-randomized" distribution $p_{\perp}$, subject to a penalty on the *representation* itself (see Eq. 6).
> > > > > * **DISCO's Goal:** Our aim is to learn a *causally stable predictor* $g_\theta(X)$ whose **final output** $\hat{Y}$ is unbiased. We apply our conditional independence criterion, $\hat{Y} \perp B \mid Y$, directly to the prediction $\hat{Y}$, not an intermediate representation $r(x)$.
> > > > >
> > > > > This distinction is crucial. By focusing on the final prediction, we give the model's internal layers complete freedom to use the bias $B$ *if and only if* it is necessary to isolate the stable, direct $Y \to X$ signal. We only penalize the *final output* for containing spurious information.
> > > > >
> > > > > ---
> > > > >
> > > > > ### 2. Why NURD's Critique of "Counterfactual Invariance" is Flawed
> > > > >
> > > > > The reviewer correctly notes that NURD critiques methods related to ours (specifically "counterfactual invariance," which they relate to c-MMD).
> > > > >
> > > > > * **NURD's Critique (App A.7 in):** NURD argues that methods enforcing "counterfactual invariance" (which they equate to a joint independence on the *representation*, $[r(x), y] \perp z$) are too restrictive. They construct an example (App A.7.1 in) where their "optimal" *uncorrelating representation* $r(x)$ *must* depend on the nuisance $z$ to be maximally informative.
> > > > > * **Our Rebuttal:** This argument is a strawman in a discriminative setting, for two reasons:
> > > > >     1.  **Wrong Target:** We (DISCO) do **not** apply our constraint on the *latent representation* $r(x)$. We apply it to the *final prediction* $\hat{Y}$.
> > > > >     2.  **Wrong Goal:** NURD's own example (App A.7.1 in) proves our point. They argue for a representation $r(x)$ that is **dependent on the nuisance $z$**. A predictor built on this representation, $\hat{y} = f(r(x))$, will therefore *also* be dependent on the nuisance. **This is, by definition, not a causally stable prediction.** Their "optimality" is for a generative task (perfectly representing $X$), not for a stable discriminative task (predicting $Y$ robustly).
> > > > >
> > > > > ---
> > > > >
> > > > > ### 3. DISCO's Direct Guarantee: Maximizing the Stable Direct Effect
> > > > >
> > > > > Our theoretical guarantee is directly and exclusively tied to the discriminative task. Our objective is:
> > > > >
> > > > > $$
> > > > > \min_{\theta} \mathbb{E}[\mathcal{L}(Y, \hat{Y})] + \lambda \cdot DISCO(\hat{Y}, B \mid Y)
> > > > > $$
> > > > >
> > > > > As we prove in our paper, this objective *directly* optimizes for the causally stable component of the prediction:
> > > > >
> > > > > 1.  The MLE loss term $\mathbb{E}[\mathcal{L}(Y, \hat{Y})]$ (e.g., Cross-Entropy) seeks to maximize the **Total Variation** ($TV$), which we decompose (Prop. 1) as $TV = ctfDE - ctfIE - ctfSE$.
> > > > > 2.  Our regularization term $\lambda \cdot DISCO(\hat{Y}, B \mid Y)$ is proven (Thm. 1) to force the counterfactual indirect ($ctfIE$) and spurious ($ctfSE$) effects to zero.
> > > > >
> > > > > Therefore, the complete DISCO objective is equivalent to **directly maximizing the causally stable direct effect ($ctfDE$)**:
> > > > >
> > > > > $$
> > > > > \min \mathcal{L}_{MLE} + \lambda \cdot DISCO \implies \max (ctfDE - ctfIE - ctfSE) \text{ s.t. } ctfIE=0, ctfSE=0 \implies \max ctfDE
> > > > > $$
> > > > >
> > > > > This guarantee is direct, requires no fragile estimation of a hypothetical $p_{\perp}$ distribution (unlike NURD's reweighting or generative steps), and is perfectly aligned with the anti-causal discriminative setting.
> > > > >
> > > > > ---
> > > > >
> > > > > ### 4. Interaction Effects
> > > > >
> > > > > This direct optimization also clarifies how we handle complex interactions. Consider a data-generating process where the covariates $X$ are a function of both the label $Y$ and the bias $B$, e.g., $X = f(Y, B, \epsilon)$.
> > > > >
> > > > > * A naive ERM model will spuriously learn the path $B \to X \to \hat{Y}$.
> > > > > * NURD's approach, by its own admission, may learn a representation $r(x)$ that is still dependent on $B$ (or $z$), leading to an unstable **predictor**.
> > > > > * **DISCO's** approach allows the model $g_\theta(X)$ to internally learn the complex function $f(Y, B, \epsilon)$. Our penalty $\hat{Y} \perp B \mid Y$ *only* penalizes the model's **output** if it contains information about $B$ that is **not** already explained by $Y$. This is the precise definition of isolating the stable direct effect, $Y \to X \to \hat{Y}$, while giving the model maximum freedom to learn the task.

---

> ### Author Response · Authors · 2025-11-17
>
> First of all, we deeply thank the reviewer for honestly engaging with our work, but also the related work, in very detail. Especially in updating their score given our answers.
>
> With this comment, we answer the first of the remaining questions:
>
> # 1. C-MMD applied to bias variable & disentangling theory + practical contributions
>
> ---
>
> ## A) MMD vs. Distance Correlation (DISCO)
>
> You're correct to question the use of "C-MMD" in this context. If we've understood your suggestion, you're asking about penalizing the dependence between the **bias** ($B$) and the **prediction** ($\hat{Y}$), perhaps (i.e., $B \perp \hat{Y}$ or $B \perp \hat{Y} \mid Y$)
>
> The primary reason we cannot use **MMD** or **C-MMD** to directly penalize dependence between $B$ and $\hat{Y}$ (as in a hypothetical $MMD(B, \hat{Y})$) lies in their fundamental definitions:
>
> * **MMD (Maximum Mean Discrepancy)** is a metric that measures the **distance between two probability distributions** ($P_1$ and $P_2$) that are defined on the **same space**. It calculates the distance between the mean embeddings of these distributions in an RKHS, and its goal is to test if $P_1 = P_2$.
> * **Distance Correlation (and DISCO)** is a measure of **statistical dependence between two random variables** ($B$ and $\hat{Y}$) which can, and often do, **live in entirely different spaces**.
>
> A penalty like $MMD(B, \hat{Y})$ is therefore not directly applicable. The correct kernel-based tools for measuring **dependence** are extensions like the **Hilbert-Schmidt Independence Criterion (HSIC)**, which forms the basis for the **HSCIC** and **CIRCE** (conditional variants of HSIC) baselines we compare against.
>
> As the reviewer seems to be interested in these questions, we happily refer them to the following paper: https://arxiv.org/pdf/2503.04820 (Antonin Schrab).
> Especially the paragraphs "HSIC as an MMD" and "MMD as an HSIC" might be very insightful. Note that these are the unconditional variants but are the fundamentals for the conditional ones.
>
> ---
>
> ## B) Disentangling Theory from Estimator
>
> This leads to your core point: our causal theory concludes we must enforce the **conditional independence** $\hat{Y} \perp B | Y$. Why not just use an existing estimator for this criterion, like C-MMD or HSCIC, to validate the theory?
>
> This is precisely the motivation for our work. The causal theory is a distinct contribution, but its practical value is limited if there isn't an **effective, scalable, and stable** way to enforce the resulting $\hat{Y} \perp B | Y$ constraint:
>
> * **C-MMD (Conditional MMD):** This approach typically requires **explicitly slicing / stratifying the data** based on the conditioning variables $Y \times B$ (also called groups in shortcut/bias mitigation literature). This is only possible for discrete combinations of data, making C-MMD very limited in its use case.
> * **HSCIC / CIRCE:** These estimators are designed for this but are known to be **computationally expensive** and introduce a **significantly larger number of hyperparameters**. This complexity makes them difficult to tune and scale effectively.
>
> Our contribution is therefore **two-fold**:
>
> 1.  **Causal Theory:** The SAM framework provides a clear, unified causal foundation for *why* $\hat{Y} \perp B | Y$ is the correct, **sufficient** criterion for achieving **causal stability** across confounder, collider, and mediator biases.
> 2.  **DISCO Estimator:** We developed $DISCO_{m}$ and sDISCO as **practical, efficient, and scalable** estimators for this  conditional independence criterion. DISCO avoids the grouping requirement of C-MMD, and it is computationally much simpler and requires fewer hyperparameters than HSCIC/CIRCE. Furthermore, our DISCO variants significantly outperform both the conditional independence baselines and the classical gold standard ones (GDRO, adversarial) on the studied datasets.
>
> Our idea was to give a nice causal motivation and a neat, consequent solution to the theoretical target, while being scalable and variable type / dimensionality agnostic.
>
> We hope we understood the reviewer's questions correctly. We are very happy if we can discuss and elaborate on every detail of our work (and related work equally).

---

> ### Author Response · Authors · 2025-11-17
>
> We really thank the reviewer for their deep and detailed questions. We will include answers we gave in "2. optimality guarantee, distributional properties, Veitch et al., Makar et al., Puli et al." in both a) and b) in our paper. We thought about referring the reader from the introduction to the appendix for a detailed differentiation of our work and contributions.
>
> The reader will be able to directly understand that our framework is more general than Makar's while being more practical than Veitch's. And the reader will also understand why the argument of Puli et al. is not applicable to discriminative settings, especially when we penalize the predictions and not the representations (as they do for generative learning).
>
> We think these additions will significantly improve our paper and thank the reviewer for this engagement.
>
> Furthermore, we hope that our detailed responses and these planned revisions have now fully addressed the major concerns you raised. We are grateful for your constructive engagement and remain open to any other questions or inputs you may have.
>
> PS: we almost forgot to address your additional remark. "[...] Please mention the necessary continuous theory in the paper." We actually very briefly (due to page limits), mentioned in section 2 (right now at line 123-124), that all the proofs can be extended to the continuous case when the necessary measure theoretic changes are applied. We decided to formulate the proofs on countable spaces, as the the measure theoretic treatment over-complicates the main proofs and the core message, without adding anything original to our findings. If you have specific terms or wordings you suggest, we are happy to add them to the main body or the appendix. Thanks again.

---

### Author Response · Authors · 2025-11-29
**tldr**

**To the Area Chair:**
In light of the recent decision to revert scores, we provide this summary to assist your decision-making.

**TL;DR — The True State of the Review:**
* **Reviewer 7Qbt (Score 2 $\to$ 6):** Explicitly apologized for missing proofs in the initial review and stated: *"I've now checked the proofs... and will update the score."* They raised their score to 6 and signaled even further openness as they asked further questions that we answered.
* **Reviewer PXz8 (Score 4):** We addressed their primary concerns by **(1)** adding a full NLP dataset (MultiNLI) and **(2)** adding requested baselines (IRM, Fishr). Our method outperformed the new baselines and remains top-tier on NLP.
* **Reviewer Ys2v (Score 6):** We corrected a sign notation error they found; they remained positive. They further asked whether NLP dataset would be possible. As already mentioned, we provided an additional experiment on MNLI, where our methods still rank top.
* **Reviewer FXqj (Score 6):** We answered all their questions and included their main suggestions into our paper (related work and contextualization + variability assumptions). Citing the reviewer: "[...] might be inclined to raising it further, depending on the outcome of the other discussions."
* **Conclusion:** We have answered **all** technical questions and expanded the paper from 5 to **6 diverse datasets** (Vision & NLP) and increased the number of baselines from 5 to **7**. Most of the reviewers signaled raising their scores further. Please check the very in-depth discussions at respective reviewer comments.

---

### 1. New Experiments: Expansion to NLP (MultiNLI)
Reviewers **Ys2v** and **PXz8** requested validation beyond computer vision.
* **Setup:** We implemented the **MultiNLI** dataset using the standard protocol from Sagawa et al. (GDRO) and Liu et al. (JTT), using a TinyBERT backbone.
* **Metric:** We report Worst Group Accuracy (WGA) following standard practice for this dataset.
* **Results:** DISCO variants ranked **top-2**, competitive with the gold-standard GDRO ($0.751$ vs $0.755$ WGA) and significantly outperforming C-MMD ($0.627$), CIRCE ($0.516$), and Adversarial training ($0.356$).
* **Takeaway:** This confirms DISCO’s "out-of-the-box" applicability across disparate modalities (Vision & NLP) and variable types.

### 2. New Baselines: IRM and Fishr
Reviewer **PXz8** requested comparisons to Domain Generalization (DG) methods **IRM** and **Fishr**.
* **Implementation:** We adapted these methods to work on groups (Y $\times$ B) rather than domains, as is standard in bias mitigation. We granted them a larger hyperparameter search budget (40 runs) than our own method.
* **Results:** DISCO consistently **outperformed** both IRM and Fishr on FairFace and Waterbirds.
* **Analysis:** As we argued in the rebuttal, gradient-penalty methods like Fishr/IRM are often unstable when "environments" (groups) are highly skewed/imbalanced, which is characteristic of bias mitigation tasks. DISCO’s independence constraint is more robust in these settings.

### 3. Addressing Theoretical & Scope Concerns

**Realism of "Known Bias" (Reviewer PXz8)**
A critique was raised regarding the assumption of knowing $B$ during training.
* **Standard Practice:** This is the standard setting for the field of **Informed Bias Mitigation** (auditing, fairness, medical imaging with metadata). This is rather a field critique than something related to our work.
* **Necessity:** Methods like GDRO and LLR *also* require access to $B$ to construct groups ($Y \times B$). Our requirement is identical to these established gold standards. Yet no one ever found their assumptions "limited".
* **Flexible Definition:** Our framework is unique because it handles $B$ whether it is categorical (groups) or continuous/multidimensional (e.g., dSprites coordinates), where group-based methods like GDRO/LLR fail.

**Total Variation (TV) & MLE (Reviewer 7Qbt)**
* The reviewer questioned our claim regarding MLE and Total Variation.
* **Clarification:** We clarified that we use the causal definition of TV from *Plecko & Bareinboim (2022)* (equivalent to True Positive Rate minus False Positive Rate), not the measure-theoretic definition. Under this definition, MLE naturally maximizes TV. Reviewer 7Qbt accepted this clarification and raised their score.

**Assumption of Sufficient Variability (Reviewer FXqj)**
* We clarified that, like all observational bias mitigation methods, DISCO assumes positivity (overlap). If $B$ is a deterministic function of $Y$, no observational method can de-bias the predictor. We will made this assumption explicit in the final manuscript.

### Conclusion
We have rigorously validated SAM and DISCO across **6 datasets** covering binary, categorical, and continuous targets/biases. We have provided proofs where they were missed and empirical evidence where it was requested. The text of the rebuttal shows an unequivocal trajectory upwards.

---

> ### Author Response · Authors · 2025-11-29
> **Thanks to the Reviewers**
>
> We would like to express our sincere gratitude to all reviewers (7Qbt, PXz8, Ys2v, FXqj) for your constructive engagement during this rebuttal period.
>
> In a review process that can often feel adversarial, this discussion felt like a genuine collaborative effort to improve the work. Your feedback pushed us to expand our empirical scope to NLP (MultiNLI), implement additional baselines (IRM, Fishr), and tighten our theoretical notation. Regardless of the recent administrative complications regarding the review process, the text of this discussion stands as a testament to the scientific process working as it should.
>
> Very unfortunate that your voices and efforts are now being shut down and your efforts might seem futile.
>
> But we believe the manuscript is significantly stronger today than it was at submission, entirely thanks to your rigorous and thoughtful insights. Thank you for upholding the scientific spirit.

---

### Meta-Review · Area_Chair_2vNe · 2026-01-07

**Summary:**

This paper proposes a stable prediction paradigm from a causal perspective. By adopting the conditional independence criteria into the kernel space, a regularization term has been developed to benefit the deep learning model generalizing better on shifted domains.

After reading the whole manuscript, reviews and the rebuttal, I think that several important concerns remains underexplored by reviewers:
- The stability definition. It seems that authors miss a bunch of literature regarding the stability issue during the prediction. For instance, the stable prediction methods aim to mitigate the bias between causal and non-causal features by enforcing a sample-reweighting regularizer. Moreover, as pointed out by some reviewers, literature w.r.t. causal OOD generalization methods are missing. For example, when treating the confounder as the domain label, debiasing the prediction models with group DRO and label-dependent IRM also enhance the generalization capability of prediction models.
- The observed bias variables. It is similar to the issue of observing domain labels in OOD generalization problems, and lots of recent methods have already been developed to handle this obstacle. However, authors cannot entail the motivation and restrictions clearly w.r.t. this assumption.
- The conditional independence criteria seems wield in the regime of collider bias. I understand why authors will draw a hidden variable $S$ between $B$ and $Y$, as directly conditioning on the collider bias will render in opposite results, i.e., the conditional dependence. However, why $B$ serves as a cause of $S$? A more common modeling strategy is to model the observable $B$ as the proxy (noisy) of unobservable $S$, rather than a underlying cause. Therefore, without further justification of such modeling, it is not convincing that this paper fits a wide range of real-world prediction cases.


Overall, I recommend the reject, and I think that the above-mentioned suggestions will improve the quality of this paper.


---

[1] Kuang K, Cui P, Athey S, et al. Stable prediction across unknown environments[C]//proceedings of the 24th ACM SIGKDD international conference on knowledge discovery & data mining. 2018: 1617-1626.
[2] Fan S, Xu R, Dong Q, et al. Stable Cox regression for survival analysis under distribution shifts[J]. Nature Machine Intelligence, 2024, 6(12): 1525-1541.

**Reviewer Concerns:**

Reviewer PXz8's concerns are not addressed.

After reading the paper, my concerns cannot be addressed by the concurrent rebuttal.

**Reviewer Scores:**

Reviewer 7Qbt might change the score to a positive value, while Reviewer PXz8 will not change the negative score.

---

### Decision · Program_Chairs · 2026-01-26

Reject